# Synthesis, Properties, and Applications of Metal Organic Frameworks Supported on Graphene Oxide

Sahar Altegani Ebrahim Naser [1],*, Kassim O. Badmus [2],* and Lindiwe Khotseng [1]

[1] Chemistry Department, University of the Western Cape, Cape Town 7535, South Africa; lkhotseng@uwc.ac.za
[2] Industrial Chemistry Department, First Technical University, Ibadan 200255, Nigeria
* Correspondence: 4116280@myuwc.ac.za (S.A.E.N.); kassim.badmus@tech-u.edu.ng (K.O.B.)

**Abstract:** Nanotechnology is one of the most active research fields in materials science. Metal-organic frameworks (MOFs) have the benefits of having a sizable specific surface area, extremely high porosity, changeable pore size, post-synthesis modification, and extreme thermal stability. Graphene oxide (GO) has attracted significant research interest due to its similar surface area to MOFs. Furthermore, oxygen-containing groups presented in graphene oxide offer the unique processing and handling advantages of amphiphilicity and dispersion in water. MOF-based GO has recently attracted attention due to its resemblance to metal ions and organic binding linkers. It has sparked great interest in the past few years due to its distinct characteristics and higher performance compared to MOFs or GO alone. This review aims to describe the most current developments in this topic for researchers. An attempt has been made to provide a synopsis review of recent research on MOFs/GO composites' properties, synthesis techniques, advantages and challenges, and different applications, including supercapacitors, gas separation and storage, water purification, sensing, catalysis, and biomedical.

**Keywords:** metal-organic frameworks (MOFs); nanomaterial; photocatalysis; electrocatalysis adsorption; wastewater; carbonaceous materials; thermal stability





## 1. Introduction

The study of nanotechnology is one of the most active areas in materials science. Nanomaterials have a great potential to alter any product's design, manufacturing, and properties [1]. The qualities of a material, such as its electric, optical, antibacterial, mechanical, and magnetic capabilities, are significantly altered when the scale is reduced from macro- to nano-size [2]. The main reason is a rise in the surface area-to-volume ratio and surface reactivity, which influence the substance's chemical and physical properties [3]. One of the criteria used to categorize nanomaterials is the kind of material they consist of. They can roughly be categorized into carbon-based nanomaterials (metal-organic frameworks (MOFs), graphene oxide (GO), carbon nanotubes (CNTs), and activated carbon (AC)), polymeric nanomaterials, metallic nanomaterials, biomolecule-derived nanoparticles, and ceramic-based nanomaterials, depending on the source material [4,5]. MOF nanostructures are currently receiving a lot of interest from both the academic and industrial worlds because of their novel contributions to medicine, remote sensing, wastewater treatment, air decontamination, and renewable energy sources [6].

The application of various materials in numerous fields is the focus of the fascinating and significant interdisciplinary field known as materials science research [7]. The quantity of investigations in materials science has continuously increased since its exceptional evolution, which began in 1995 when Yaghi and colleagues formed the MOFs, a specific category of porous material made up of inorganic metal ions and organic linkers [8]. MOFs are the coordination of metal ions/clusters with organic bridging linkers that serve as the inorganic and organic molecular building blocks. Furthermore, they are a remarkable class of crystalline, porous, and hybrid materials that can be made under mild conditions using

mechanochemical, microwave, electrochemical, ultrasonic, and solvothermal methods [9]. Figure 1 illustrates the metal ions acting as nodes, which are a cage shape connected to the linker's arms to synthesize a repeating structure. MOFs have a colossal internal surface area because of their hollow nature [10,11]. Numerous reports have been conducted on the diverse shapes, such as amorphous, cubic, and hexagonal spherical. Additionally, two-dimensional (2D) and three-dimensional (3D) MOFs produced by heterojunctions were reported as additional MOFs morphologies from the post-synthesis approach [12].

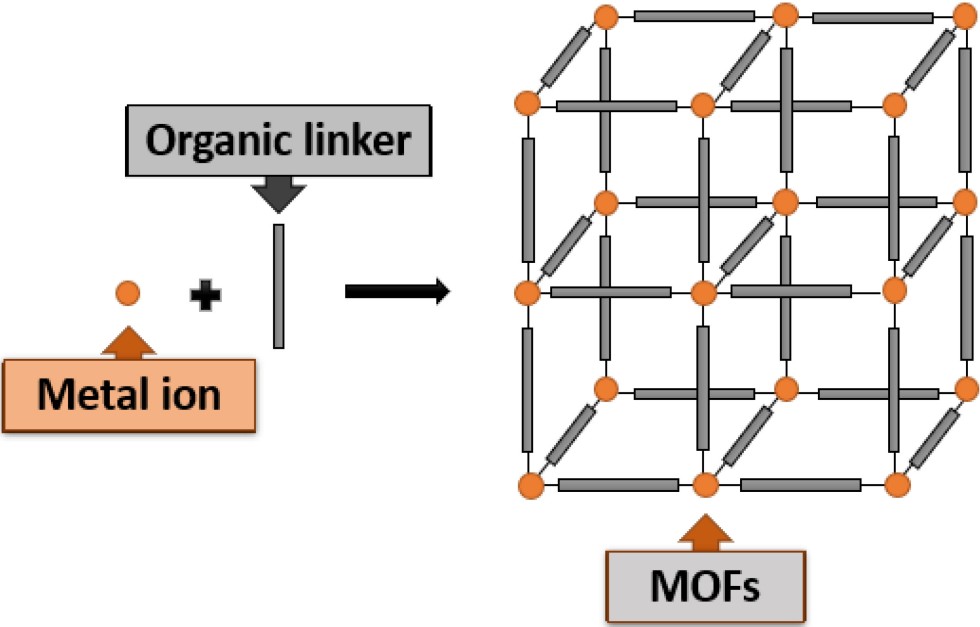

**Figure 1.** Schematic diagram of the synthesis procedure for MOFs.

The great specific surface area (a Langmuir surface area greater than 10,000 m$^2$ g$^{-1}$), extremely high porosity, changeable pore size (90% or more free volume), post-synthesis modification, high-order crystalline substances, high modularity, and significant thermal and chemical stability are all advantages of MOFs [13,14]. MOFs with constant porosity and adaptable chemical properties can be produced for differing purposes [15]. In light of their numerous applications in gas separation and storage, electric, optical, and magnetic materials, remote catalysis, sensing, biomedicine, battery applications, and other fields, MOFs stand out as the best platform for creating new multifunctional materials [16–18]. The synthesis of fine chemicals is essential to the chemical industry, which widely uses MOFs as heterogeneous catalysts [19]. Active sites are unquestionably necessary for catalytic applications, and the kinds of origin elements that make up these sites are crucial, as indicated in Scheme 1. In pristine MOFs, the catalytically active sites originate at the unsaturated metallic Lewis acid, and the coordinating organic linkers act as the base sites [20].

Frequently, MOFs have been constructed from functional molecular catalysts (Schiff-base complexes, metalloporphyrins, etc.), which act as building blocks for connected catalytic reactions. Nevertheless, the limited number of active sites observed in pristine MOFs severely restricts the applicable range of reactions and performance enhancements [20]. Therefore, MOFs are a suitable form of heterogeneous catalysts because of their potential for selectivity [21].

Despite having a controllable structure and composition, MOFs have several essential limitations and challenges, including low electrical conductivity and occasionally insufficient chemical, mechanical, or thermal stability, which restricts their use in the electrochemical area [22]. A fundamental problem with many (but not all) MOFs is their lack of structural stability under reaction conditions and their inability to be reactivated by

heat treatment [23]. However, because electrons and poles are not separated well enough, significantly pure MOFs are severely constrained [24]. Hence, there are numerous constraints on specific approaches to produce MOFs employing soluble metal precursors. The dissolved metal ions react with the bridging linkers at an accelerating pace, leading to side reactions and the consequential production of materials that are often low in purity and high in disorder [25]. Solvent recovery may become complicated when ionic metal salts quickly succumb to dissolution, leading to diffusion and oxidation issues [26]. It is hard to control MOFs' crystal development during synthesis because conventional soluble metal precursors are challenging to work with and exploit in space and time [27,28]. Direct synthesis of MOFs from metal ions and ligands is not easy to form due to problems such as incorrect reaction temperatures, the unavailability of suitable solvents, and restricted reactant concentrations [29]. Another critical disadvantage of MOFs is their void space, which makes it complex to retain tiny molecules such as medicinal substances in an ambient environment [30].

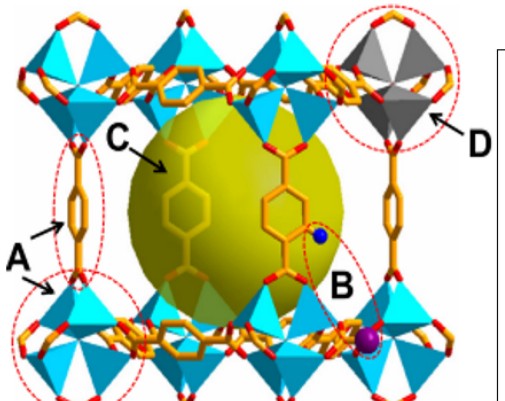

Origin of the catalytically active site in materials based on MOFs: (A) pristine MOFs contain sites on metal clusters and linkers; (B) modified grafted sites in MOFs; (C) MOFs composites with encapsulated sites; (D) upon thermal or chemical conversion, sites are produced in MOFs derivatives.

**Scheme 1.** Catalytically active site origin in MOF-based materials. Reproduced with permission from [20], copyright 2023 Elsevier.

The MOF drawbacks were solved by exploring a rational design of 2D-based catalysts. Novoselov et al. For the first time in 2004, Novoselov and his colleagues explored graphene as a typical advanced 2D material. Later, transition-metal dichalcogenides (TMDs), such as graphene, phosphorene, and germanene, have drawn increasing interest [31]. Graphene-based materials have recently opened up new potential applications due to their distinct structure, mechanical properties, electrochemical properties, low toxicity, thermal properties, and superior electronic properties. More recently, several reports have verified that adding graphene-based materials to other materials can lead to unexpected increases in electronic conductivity and stability. Considering that the parent materials have complementary properties, it is reasonable to ask whether composites made from the combination of graphene-based materials and MOFs have exceptional qualities in these circumstances. Fundamentally, the concept of hybridizing graphene derivatives with MOFs appeared and quickly gained popularity [32]. Fortunately, the term of combining MOFs with other materials to synthesize a MOF-based composite that can combine the benefits of both parent materials has recently been proposed [33]. Research efforts have concentrated on assortment techniques, such as MOF-based composites, linker functionalization, post-synthetic modifications, inorganic building unit connectivity, and dimensionality [34]. The purpose of MOF-based composites created by combining MOFs with other functional materials is to increase the surface area, separation ease, and adsorption capacity. The term "derived MOFs" describes raw materials that have been directly carbonized [35]. Additionally, much work has been carried out in producing metal-carbon porous MOFs with stable structures and improved conductivity [36]. A testament to the exceptional combination of graphene's outstanding features is the meteoric emergence of graphene-based MOFs. A theoretical surface area of 2630 $m^2/g$ and excellent thermal ($5 \times 10^3$ W/mK) and electrical

$(2 \times 10^5 \text{ cm}^2/(\text{Vs}))$ conductivity are only a few of graphene's record-breaking physical and chemical characteristics. The oxygen-functional groups obtained on the graphene surface make GO the most significant graphene derivative [37]. MOF/GO-based composites are particularly attractive due to their potential for achieving synergistic effects between the GO and MOFs. This potential is due to GO's advantageous properties, such as high electron mobility, mechanical stability, high thermal and electrical conductivity, etc., and porous solids (MOFs) have preferable properties, such as catalytic activity, controlled porosity, selectivity, etc. [38,39]. Therefore, a combination of GO and MOFs can lead to addressing the essential shortcomings of the MOFs [40].

Since graphene significantly differs from graphite in terms of most of its properties, it has remarkable applications in many areas, including sensors, photovoltaics, energy storage, and electronics. Therefore, the discovery of graphene has sparked interest in various contemporary nanomaterial technology fields [41]. Graphene is composed of carbon and resembles a beehive shape [42]. The graphene structure consists of a separate layer of $sp^2$-hybridized carbon atoms, as depicted in Figure 2. With a sheet thickness of 0.34 nm, it is a two-dimensional carbonaceous material and the thinnest and strongest nanomaterial [43]. Graphene has been demonstrated to possess several desirable qualities, including great mechanical strength ($42 \text{ Nm}^{-1}$, 1.0 TPa for Young's modulus, and 130.5 GPa for intrinsic tensile strength), electrical conductivity ($25 \text{ m}^2\text{V}^{-1} \text{ s}^{-1}$ of high electron mobility), the capacity to form molecular barriers, and other outstanding characteristics [44].

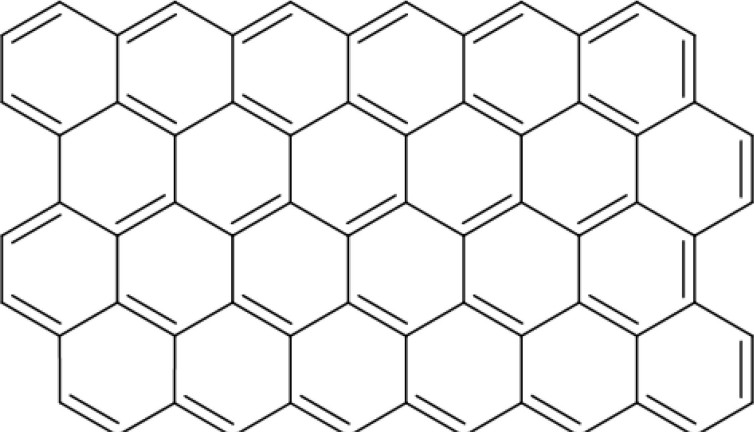

**Figure 2.** The structure of graphene.

Graphene can be oxidized and exfoliated to produce GO (Figure 3), a hydrophilic carbon-based sheet with many oxygenated functional groups. It has functional groups with oxygen-based atoms such as hydroxyl (-OH), carboxylic acid (-COOH), alkoxy (C-O-C), carbonyl (C=O), and others [45,46]. GO has attracted significant research interest because it contains a large surface area ($2630 \text{ m}^2/\text{g}$), similar to graphene, and the unique processing and handling benefits of amphiphilicity and dispersibility in water provided by the present oxygen-containing groups [47]. The applications of GO in electronics, conductive films, electrode materials, and nanocomposites have effectively been published [48]. The most extensive method for producing GO is the Hummers method, which uses concentrated sulfuric acid [49] as a result of its capable thermal and chemical stability, lightweight, elastic strength, high tensile strength, and exceptional optical and electrical properties [50].

**Figure 3.** Oxidation of graphene to form graphene oxide.

## 2. The Properties of Metal-Organic Frameworks/Graphene Oxide (MOFs/GO)

The combination concept of GO material and MOF substrates was suggested to address the respective drawbacks of GO and MOFs. The composite membrane has a bifunctional characteristic that allows it to interact with metal ions in MOFs due to the presence of epoxy and hydroxyl functional groups on either side of the GO lamellae [51].

Using 2-methylimidazole as a reasonably priced ligand and water as a solvent, Hong et al. [52] successfully synthesized Ni-Co-MOF/GO at a low temperature. Graphene oxide, which has a two-dimensional (2D) structure, and MOF, which has a three-dimensional (3D) structure with hierarchical pores and high stability, synergistically work together to improve properties such as the large internal surface area. At a current density of 1 A $g^{-1}$, the specific capacities of Ni-Co-MOF and Ni-Co-MOF/GO were 120.0 and 230.9 F $g^{-1}$, respectively. The addition of graphene oxide to Ni-Co-MOF resulted in the Ni-Co-MOF/GO electrode having excellent electrochemical properties. Moreover, the Ni-Co-MOF/GO electrode, employed as a developing material for supercapacitors, demonstrated outstanding electrochemical capabilities due to the combination of GO and Ni-Co-MOF. Lastly, the Ni-Co-MOF/GO exhibited high energy storage qualities due to the 3D nanostructures of Ni-Co-MOF with GO [52].

In a different work, Kumar et al. [53] used a grinding approach to synthesize MOF-5@GO nanocomposites in a single-pot process. Due to its instability in water, the MOF-5 has a distinctive issue, and the combination of the MOF-5 and GO is chiefly required to address its flaws. Based on this, the interactions between GO and MOF-5 composites improve the MOF-5 framework's hydrophobic stability. Consequently, the MOF-5@GO nanocomposite enables interactions between oxygenated functional groups of GO and open sites of the zinc cluster of MOF-5. As a result, the maintenance of crystallinity boosts MOF-5's conductivity and improves its adsorption characteristics [53].

In a different study, Yang et al. [54] used the in situ growing method to prepare the HKUST-1@GO composite. The HKUST-1 particles in the HKUST-1@GO hybrid composites spread uniformly on the GO layers due to the strong bonds between the oxygen groups contained in GO and the HKUST-1 metal clusters, instead of physical mixing with GO (Figure 4). This structure allowed the HKUST-1@GO composites to fully use the exclusive qualities and functions of the GO layers by efficiently inhibiting the agglomeration of the layers. Moreover, the roughness of the surface reduces the hybrid membrane's sensitivity to fouling. It is clear from the results that membranes modified with MOF@GO work satisfactorily during the water purification operation, proving that HKUST-1@GO is the best filler for creating improved ultrafiltration membranes. When HKUST-1 and GO are combined, as opposed to only adding GO, the shortcomings of GO can be successfully solved, allowing the properties of GO to be fully utilized [54].

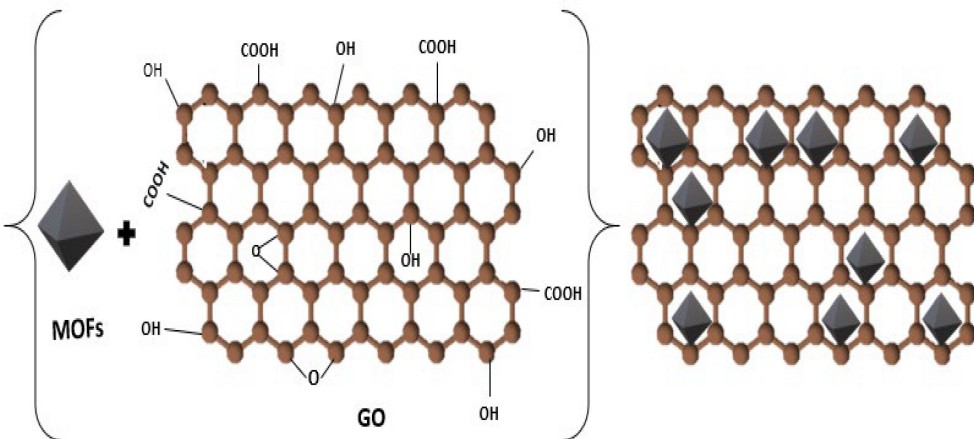

**Figure 4.** Formation of MOFs over the GO layer.

Using a hydrothermal technique, Heu et al. [55] generated photocatalytic nanofiltration (NF) membranes with improved flux and antifouling capabilities from a layered in situ nanocomposite of MOFs (UiO-66) and GO (UiO-66_GO) over a polyamide NF membrane. In situ growth of UiO-66 particles on a (2D) GO sheet revealed that the GO oxygen functional groups enhanced UiO-66 formation, while raising the dispersion force between UiO-66 particles and stifling their aggregation, enabling the control of their physicochemical characteristics, such as morphology, size, and structure. By making the membrane surface more hydrophilic and smoother, the UiO-66_GO nanocomposite at a loading of 15 wt% increased pure water flux by 187% compared to the pristine NF membrane flux. However, due to its high flux improvement (169%) and the UiO-66_GO nanocomposite's good stability on the membrane surface, the loading of 10 wt% of the UiO-66_GO composite was ideal. Additionally, the higher hydrophilicity, enhanced smoother surface, and the charge repulsion of the UiO-66_GO/NF-10% membrane were also associated with a higher FRR (80%) and lower Rir (20%), in contrast to the pristine NF membrane. The composite membrane also exhibited photocatalytic activity, increasing the FRR to 98% and destroying the accumulated Suwannee River Humic Acid (SRHA) [55].

Aimed at synthesizing a solid-phase microextraction (SPME) fiber for the derivatization-free extraction and analysis of nonsteroidal anti-inflammatory drugs (NSAIDs), in conjunction with gas chromatography (GC), Liu and associates [56] synthesized a composite material consisting of a zirconium-based metal-organic framework and graphene oxide (Zr-MOF@GO) using a hydrothermal method. GO is a recognized leader among nanomaterials because of its high specific surface area and abundance of functional oxygen groups, including hydroxyl and epoxy groups, and its ability to modify surfaces in many ways. Hence, adding GO to MOFs enhanced the adsorption properties of the material. The high scratch resistance of the fiber coating can successfully stop the failure of the coating and corrosion due to friction between the casing and the fiber. As a result, the Zr-MOF@GO-coated SPME-GC-FID method has possibilities for the accurate analysis of NSAIDs in water [56].

## 3. Synthesis Methods of Metal-Organic Frameworks/Graphene Oxide (MOFs/GO)

Ordinarily, the previous process of synthesizing nanomaterials derived from MOFs is regularly employed to prepare MOF/GO composites. Numerous strategies exist to produce MOFs/GO composites, including post-synthesis, in situ, solvothermal, co-precipitation, mixing, and ultrasonication, all listed in Table 1, along with their advantages and challenges [57]. The synthetic conditions significantly influence the types of interactions between MOFs and GO or reduced graphene oxide (rGO), which results in the development of numerous synthesis techniques and composite properties. The manufacturing of MOF/GO composites and MOF/rGO is identical. Moreover, in contrast to the synthesis

pathways of GO and rGO, the reaction temperature will not be considered because GO goes through a reduction when the synthesis temperature reaches or exceeds 100 °C [58].

### 3.1. Post-Synthesis Method

The first method for modifying MOFs is post-synthetic modification (PSM), which involves grafting organic functionalities and crosslinkers onto the surface of the particles of MOFs to produce functionalization materials [59]. This technique was used to prepare MOFs/GO composites. It entails preparing the MOFs beforehand and adding them to the GO, as shown in Figure 5 [60].

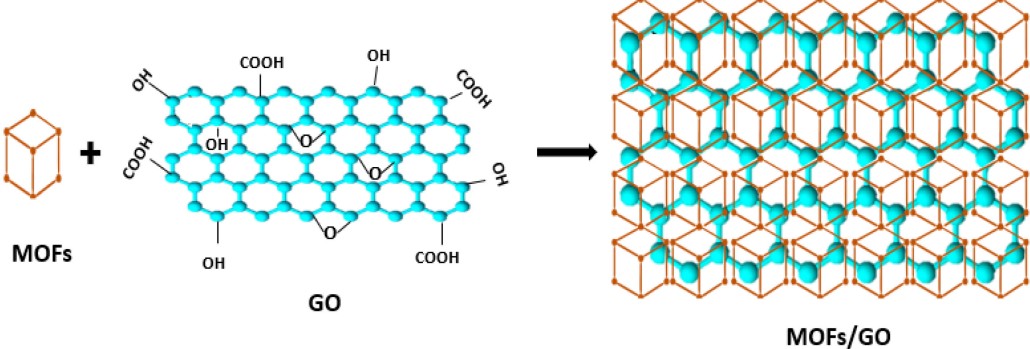

**Figure 5.** Preparation of MOF/GO via the post-synthesis method.

Through a post-synthetic approach, Karamipour et al. [61] modified TMU-10 to synthesize TMU-10@GO hybrids using GO. The outcome demonstrated that TMU-10@GO can function as the best sorbent because it contains GO and has stronger interactions, higher dispersibility, and a larger surface area than TMU-10. Additionally, the presence of GO in the TMU-10@GO hybrid markedly increased phenol's adsorption capacity and its sensitivities. Along with the potential interactions with TMU-10, adding GO also results in stronger adsorption of phenol through hydrogen bonds and π-π interactions. Moreover, the interaction between the adsorbates and accessible surface sites is the rate-limiting step, according to the pseudo-second-order model of the adsorption kinetics. Studies on the adsorption isotherm that validate the Langmuir isotherm showed that the TMU-10@GO composites have a chemical adsorption nature toward phenol adsorption. The composite TMU-10@GO was easy to use, inexpensive, performed well in adsorption, had a large surface area, reachable sites, and required little sorbent [61].

In a further study, via a post-synthetic technique, Ullah et al. [62] were able to synthesize MOF-200/GO. MOFs are defective, porous, adsorbent materials for gas separation from the perspective of industrial applications, particularly for the separation of $CO_2$. Due to the weak interfacial interactions between the adsorbent and adsorbate, MOFs have a decreased capacity to absorb $CO_2$ under low-pressure circumstances, such as atmospheric pressure (1 bar). Consequently, the absorption capacity improved in the MOF-200/GO through surface tuning, also known as surface functionalization [62].

### 3.2. Hydrothermal and Solvothermal Methods

Two of the most widely applied in situ techniques are the hydrothermal and solvothermal techniques. They are referred to as synthesis methods, where the reaction occurs at pressures greater than 1 bar and temperatures higher than the solvent boiling point. The variables support the development of crystalline products [63]. The only difference between the two methods in the synthetic pathways is the kind of solvent employed. It can either be aqueous (hydrothermal) or nonaqueous (solvothermal). Moreover, the researchers prefer this method due to its simplicity, quick processing time, and high composite yield [64].

The advancement of renewable energy systems is an intriguing possibility for electrochemical water splitting. Additionally, to reduce their overpotentials and speed up their practical application, it is strongly suggested to fabricate suitable electrocatalysts for water

splitting. Ahmed Malik et al. [65] effectively produced Ce-MOF and GO@CeMOF using solvothermal techniques, and $CeO_2$ from Ce-MOF and GO@CeMOF using calcination. The methods necessary to fabricate the GO@Ce-MOF utilizing the solvothermal approach are illustrated in Scheme 2. The oxygen evolution reaction (OER) was studied as a potential electrocatalyst using the preceding catalysts, with the lowest overpotentials, tiny Tafel slope, minimum impedance, and highest peak current density, where the electrocatalysts displayed exceptional results. The electrocatalytic activity of the calcined samples was more effective when compared to that of simple MOFs and their composites. Although, in contrast to Ce-MOF and GO@Ce-MOF, the calcined samples ($CeO_2$) exhibited accelerated catalytic activity due to the presence of exposed active sites in the structure of $CeO_2$. As a result, the appearance of GO layers, which increased the system's overall conductivity and offered more surface area with effective active sites for the catalytic reaction, GO@Ce-MOF had higher catalytic activity than Ce-MOF. These reliable, affordable, and sufficient electrocatalysts were anticipated to be the leading catalysts for OER [65].

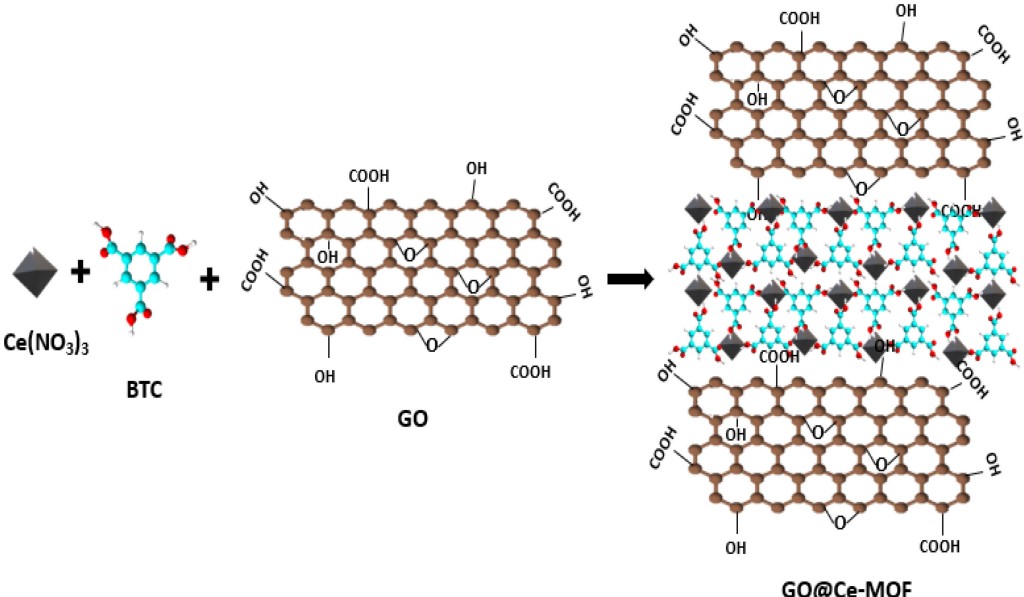

**Scheme 2.** Synthesis and calcination of the GO@Ce-MOF for the OER investigation.

Solvothermal synthesis was applied by Fallatah et al. [66] to prepare a ZIF-67@GO composite. The functionalization of adsorbent surfaces utilizes various approaches to improve the adsorption performance. Several investigations that used composites of MOFs with GO to produce modified materials did so successfully. To effectively remove mercury ($Hg^{2+}$) from the aquatic environments, ZIF-67@GO was tested as an adsorbent. According to the adsorption data, using 10 mg of composite in 50 mL of a 20 ppm mercury ($Hg^{2+}$) solution resulted in a maximum removal efficiency of 91.1%. The synthetic composite material had excellent reuse potential. Therefore, using the water-stable ZIF-67@GO composites, it was successfully possible to remove mercury ($Hg^{2+}$) from the water [66].

Huang et al. [67] synthesized the composites of GO and the HKUST-1 framework via the common solvothermal technique. The pre-prepared composites served as heterogeneous catalysts in catalytic wet peroxide oxidation (CWPO) of phenol. The GO-3/HKUST-1 composite catalyst could continue to perform satisfactorily for the catalytic degradation of phenol following three additional runs. Moreover, the GO-3/HKUST-1 composite demonstrated excellent stability with minimal copper ($Cu^{2+}$) leaching (8 ppm) under observation. The hydroxyl radical ($^\bullet$OH) mechanism is used in the catalytic degradation of phenol. Additionally, the GO/HKUST-1 composite may be a promising heterogeneous catalyst for the catalytic degradation of organic compounds. The production of hydroxyl radicals plays a similar role in the catalytic oxidation of phenol as other catalysts [67].

### 3.3. In Situ Method

In situ techniques are commonly employed to produce the MOF/rGO and MOF/GO composites. Once again, in situ methods are preferred because they are practical and require mild synthesis conditions, such as water as the solvent and low temperatures [68].

Zeolitic imidazolate frameworks (ZIFs) are an example of MOFs that offer better adaptability and hydro-stability. The coordination of N-atoms from imidazolate anions with metal ions ($Zn^{2+}$ or $Co^{2+}$) yields the porous nature of ZIFs. The ZIFs' structure is comparable to zeolites, which consist of four connected webs of tetrahedral building blocks. Reverse osmosis, ultrafiltration, microfiltration, and nanofiltration are just a few membrane separation techniques that incorporate ZIFs due to their chemical, hydrophilic, and thermal properties. Makhetha et al. [69] successfully synthesized ZIF-8@GO composites utilizing the in situ growing method. The pre-prepared ZIF-8@GO was then used as a filler in low concentrations of 0.1–0.5 wt% via phase inversion to produce polyethersulfone (PES) ultrafiltration (UF) membranes. Triethylamine (TEA) was used as a critical controller for the ZIF-8 nanoparticle sizes to have a uniform distribution at ca. 30–50 nm on the GO sheets. The hydrophilicity and surface jaggedness of the PES membranes were enhanced because of the incorporation of the ZIF-8@GO composite. Surprisingly, the ZIF-8@GO composite membranes increased the water permeation from ~28 (PES) to ~71 ($M_h$) L m$^{-2}$ h$^{-1}$ at 100 kPa. As a result, the synthesized UF composite membranes had extreme fouling resistance and maintained their recovery for at least six cycles while recovering ~90% of the water flux during high loading [69].

The design and production of novel MOFs/GO composites with exceptional performance have drawn increasing attention. The in situ synthesis method was applied by Li et al. [70] to produce an artfully formed PCN-222/GO composite by fabricating zirconium-porphyrin MOF (PCN-222) in the presence of GO with a COOH group. The working electrode was modified using this composite as a functional material. This composite, PCN-222, where GO has excellent electrical conductivity, contains abundant mesoporous channels and Zr(IV) metal sites. This composite can immobilize a significant amount of aptamer through strong π-π stacking interactions and a high affinity between the phosphate group of the aptamer and the Zr(IV) site of PCN-222 at the same time, because of the excellent electrical conductivity of GO, an abundance of mesoporous channels, and the numerous Zr(IV) metal sites of PCN-222. As a substrate, GO with the COOH group is used to support PCN-222 and increase its electrical conductibility. Meanwhile, this synthetic aptasensor exhibits excellent selectivity, good repeatability, and preferred long-term storage. Therefore, a PCN-222/GO composite-based, ultra-sensitive, electrochemical aptasensor can quantifiably detect trace chloramphenicol, with a limit of detection of 7.04 pg/mL (21.79 pmol/L) from 0.01 ng/mL to 50 ng/mL, via electrochemical impedance spectroscopy, even in fundamental samples [70].

In a different study, a GO/MOFs composite known as Ni-BTC@GO was synthesized in response to significant ecological and environmental issues occurring throughout the world. According to Li et al. [71], the in situ synthesis of Ni-BTC@GO heterostructures offered a practical method for the construction of high-performance supercapacitor electrodes by enhancing the charge capacitance of Ni-BTC and preventing GO aggregation. Here, 1,3,5-benzene tricarboxylic acid (BTC) is utilized as an organic linker to synthesize the composite due to its inexpensive advantages. Figure 6 illustrates the in situ method to prepare Ni-BTC@GO composites. Additionally, in correlation to the pure GO and Ni-BTC, the Ni-BTC@GO composites had a more stable electrolyte–electrode interface and a better electron transport pathway. The synergistic effects of the Ni-BTC framework and GO dispersion on electrochemical behavior show that Ni-BTC@GO 2 has the best performance for energy storage. Likewise, the findings show that the most significant specific capacitance is 1199 F/g at 1 A/g. After 5000 cycles at 10 A/g, Ni-BTC@GO 2 exhibited extraordinary cycling stability with a score of 84.47%. Furthermore, the constructed asymmetric capacitor shows energy densities of 40.89 Wh/kg at 800 W/kg and 24.44 Wh/kg at 7998 W/kg [71].

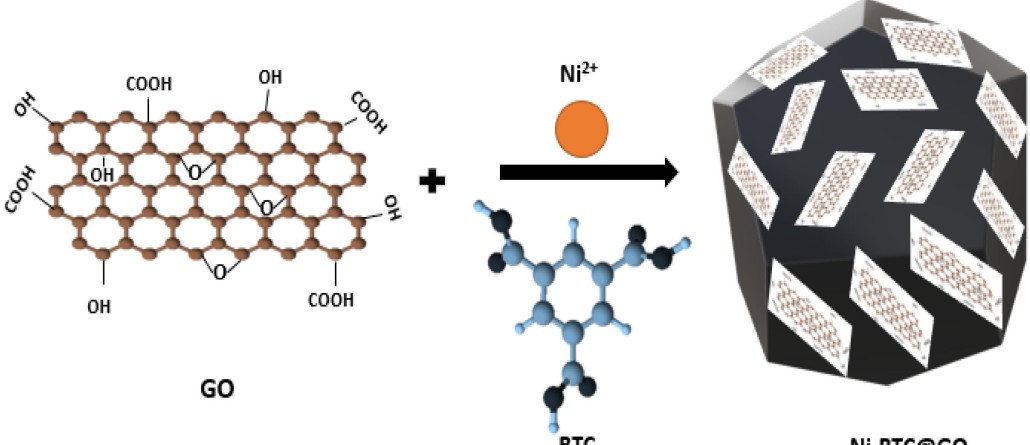

**Figure 6.** In situ method to prepare Ni-BTC@GO composites.

### 3.4. Co-Precipitation Method

In recent years, solution precipitation has been a widely used method for producing inorganic nanomaterials. The "co-precipitation method" is another name for this process because the cations simultaneously precipitate. The co-precipitation technique is easier to control, uses milder experimental conditions than hydrothermal reactions, is simple, and is faster for MOFs' production and their composites [72].

Xu et al. (79) prepared Co-ZIF-adsorbed borate ions with functionalized rGO (ZIF-67/rGO-B), conducive to lowering the fire risk of epoxy resin (EP). Firstly, ZIF-67 was loaded onto the surface of graphene using the co-precipitation method. The authors discovered that it decreased the fire danger of polymer materials and the agglomeration of rGO in composites. After that, boron ions were successfully adsorbed onto the ZIF-67 surface to produce a brand-new hybrid, known as ZIF-67/rGO-B. Moreover, ZIF-67/rGO-B was physically blended into EP, and then the fire risk of EP composites was examined using cone calorimeter tests, the limited oxygen index (LOI), and other methods. According to the combustion results, EP composites produced significantly less heat and smoke. Exceptionally, when compared to pure EP, the peak heat release rate (pHRR), total heat release (THR), and maximum value of smoke density (Ds,max) of the composite with 2 wt% ZIF-67/RGO-B were all decreased, by 65.1%, 41.1%, and 66.0%, respectively [73].

Beka and associates [74] synthesized a NiCo-MOF/rGO hybrid material by a one-step room-temperature precipitation process. A NiCo-MOF 2D nanosheet/rGO heterostructure was prepared for supercapacitor use. The MOF ultrathin nanosheets in this hybrid structure provided a sizable surface area with numerous channels for the fast mass transport of ions, while the fast electron transport provided by the conductive physical support of rGO enhanced the overall electrochemical performance. As a result, an excellent specific capacitance of 1553 F $g^{-1}$ at a current density of 1 A $g^{-1}$ was obtained by combining the benefits of rGO and NiCo-MOF nanosheets. In addition, after 5000 charge–discharge cycles, the as-synthesized hybrid material displayed an excellent cycling capacity of 83.6%. Furthermore, the completed asymmetric device exhibited magnificent energy densities of 44 W h $kg^{-1}$ at 3168 W $kg^{-1}$. Figure 7 shows the NiCo-MOF/rGO synthesis process [74].

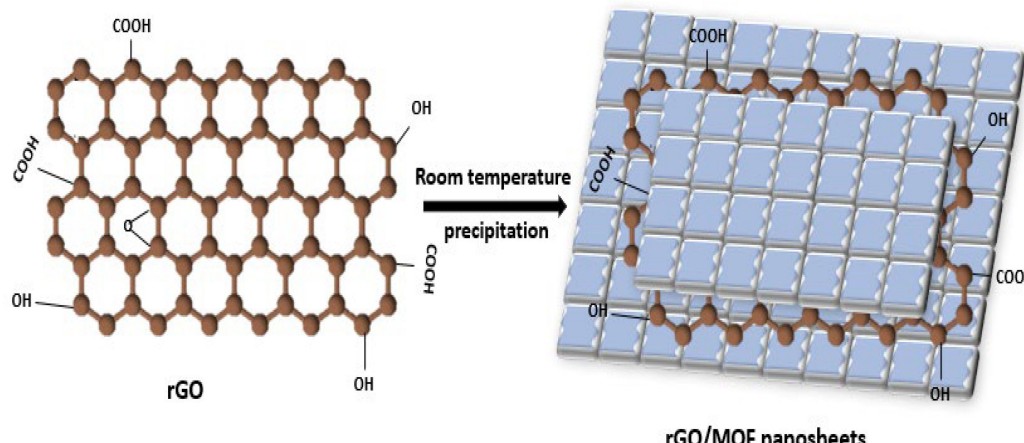

**Figure 7.** Process of synthesizing NiCo-MOF/rGO.

### 3.5. Mixing Method

Physical mixing is a quick and practical technique for synthesizing MOFs/GO nanocomposites. To produce the composite, MOFs and GO are first prepared ahead of time and then directly combined [75].

To improve the electrochemical anodic performance of the parent pure Ni-based MOF (Ni-MOF), Haroon et al. [76] prepared the Ni-MOF@rGO composite using low temperatures, linked to a representative Ni-MOF with rGO via an intermediary physical mixing method. This method was exceptional because it produced close contacts between the Ni-MOF and thermally regenerated rGO, while maintaining the MOFs' anonymity at low temperatures (300 °C). With a capacity of 385 mAhg$^{-1}$ (100 mAg$^{-1}$), which preserved its consistency for 400 charge–discharge cycles, the Ni-MOF-rGO composite, or Ni-BDC/rGO@300, had a significantly improved rate and cycling stability as compared to the pristine MOFs, which degraded to 272 mAg$^{-1}$ in just 250 cycles. The Ni-BDC/rGO@300 composite displayed a capacity of 205 mAhg$^{-1}$ at 1 Ag$^{-1}$, while the pristine Ni-MOF capacity was 113 mAhg$^{-1}$, demonstrating significant rate improvements at higher currents. Scheme 3 illustrates the general synthetic process [76].

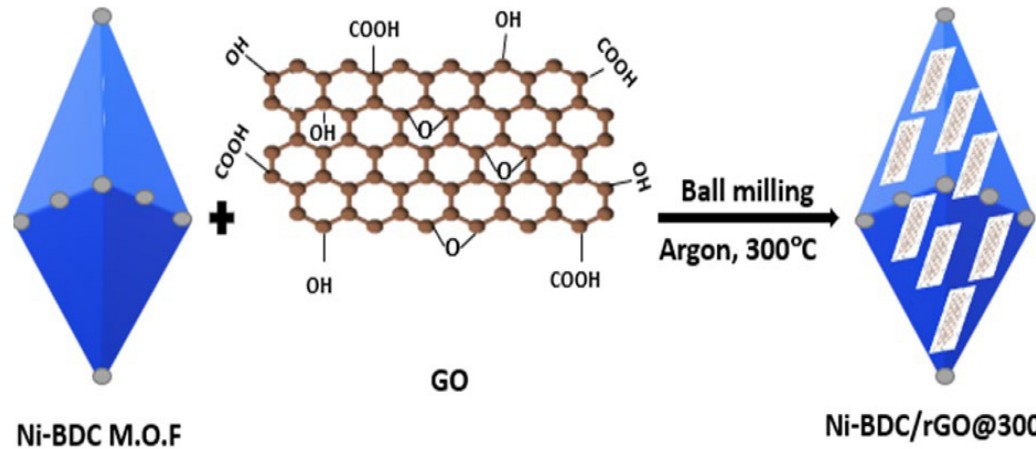

**Scheme 3.** Mixing approach for synthesis of Ni-BDC/rGO@300.

Munir et al. [77] produced a composite of copper-based MOFs with various ratios of GO for hydrogen storage applications. A well-dissolved MOF precursor was combined with GO powder to synthesize the composite. In MOF/GO composites, MOF-199 had assertive properties, and its porosity was higher than that of the parent element. According to some theories, the interaction of MOF-199 with GO functional groups (epoxy, hydroxylic, carboxylic, and sulfonic groups) promotes the development of new

pores. Up to 20% of the GO was found to have an increase in porosity. After the GO content reached 20%, functional groups on the deformed graphene layer outnumbered the number of MOF sites that could interact with one another. The experiment findings demonstrated that the MOFs/GO 20% composite material can store 6.12% of hydrogen at $-40\,^\circ$C [77].

*3.6. Ultrasonication Method*

Exceeding interest has recently been shown in the green and environmentally friendly ultrasonication method of nanomaterial synthesis [78]. This technique requires room temperatures, a non-hazardous solvent, and a quick synthesis time (less than an hour), without the requirement for harsh circumstances. Additionally, the ultrasonication method reduces the MOF's surface area and simultaneously allows an interaction between the MOF's surface area and guest molecules, thus producing impressive results. Nevertheless, despite its benefits, the ultrasonication method is constrained by things such as low stability at high temperatures and high humidity, poor performance of MOFs during the adsorption process, small pore volumes, and insufficient surface areas. These limitations can be overcome by strengthening the MOF characteristics through the functionalization of additional molecules [79].

To control the production of Cr-MOF nanoparticles with a small size, crystallinity, and large specific surface area, Cui et al. [80] developed the dendrimer PAMAM. The PAMAM/CrMOF/GO nanocomposite was prepared using a straightforward, eco-friendly, quick ultrasonication technique, demonstrating excellent water dispensability. In this way, it could have a synergetic effect on the electrochemically active area and the sensitivity of electrochemical sensing. Through a quick and effective electroreduction process, the GO material was converted into an excellent, electrically conductive reduction form that can effectively boost the conductivity of MOF nanocomposites beyond the target adsorption capacity through $\pi$-$\pi$ stacking, hydrogen bonding, and other forces. Additionally, a modified nanocomposite on a glassy carbon electrode (GCE) was used to create an electrochemical sensor. The synergistic combination of the high conductivity of ERGO and the high porosity of PAMAM/Cr-MOF resulted in nanocomposite-accelerated electron transfer, while increasing the effective specific surface area, which is why the proposed sensor displayed an enormously higher current response than the ERGO-based sensor. Under the best experimental circumstances, this technique can detect 1-OHPyr in human urine with satisfactory reproducibility and a magnificent anti-interference ability [80].

A unique copper-based metal-organic framework (Cu-MOF), immobilized on GO, has been synthesized by Firouzjaei et al. [81] using an ultrasonication technique. The results of the various investigations demonstrated that the exfoliation of GO in the Cu-MOF structure improved the GO-Cu-adsorption MOFs' capacity, in contrast with the Cu-MOF. Comparing the Cu-MOF nanoparticles with the GO-Cu-MOFs displayed an improved pore size, more active groups, negative charge, and improved surface area, leading to a 20% greater dye removal rate. The adsorption kinetic results followed the rapid adsorption process, which had a pseudo-second-order characteristic. In comparison to the 106, 117, and 142 mg/g adsorption capacities of Cu-MOF at the same temperature, the GO-Cu-MOF displayed higher adsorption capacities of 173, 251, and 262 mg/g at 25 $^\circ$C, 45 $^\circ$C, and 65 $^\circ$C. According to dye removal experiments, Methylene Blue (MB) is more likely to adhere to the GO-Cu-MOF compound at higher temperatures (65 $^\circ$C) and in acidic conditions [81].

In consideration of improving $CO_2$ separation, Yang et al. [82] successfully prepared ZIF-8@GO-based mixed-matrix membranes (MMMs) by impregnating ZIF-8-modified GO into an ethyl cellulose (EC) matrix. ZIF-8 was fully developed on the GO surface using an ultrasonic synthesis process to prepare the ZIF-8@GO formulation, as shown in Scheme 4. Compared to pristine GO sheets, ultra-porous ZIF-8-modified GO plates exhibit lasting rigidity and significantly less aggregation and folding, which results in

diminished $CO_2$ gas barrier effects. Although, compared to pristine EC membranes and GO MMMs, the apparent activation energy ratio of $N_2$ and $CO_2$ in ZIF-8@GO MMMs is higher. Since $CO_2$ and $N_2$ can be separated at a relatively lower temperature, ZIF-8@GO MMMs are more appropriate and efficient. A $CO_2$ permeability of 203.3 Barrer and a $CO_2/N_2$ selectivity of 33.4 were displayed by an EC/ZIF-8@GO membrane that contains 20 wt% hybrid fillers. These values increased over a pure EC membrane by 139% and 65%, respectively [82].

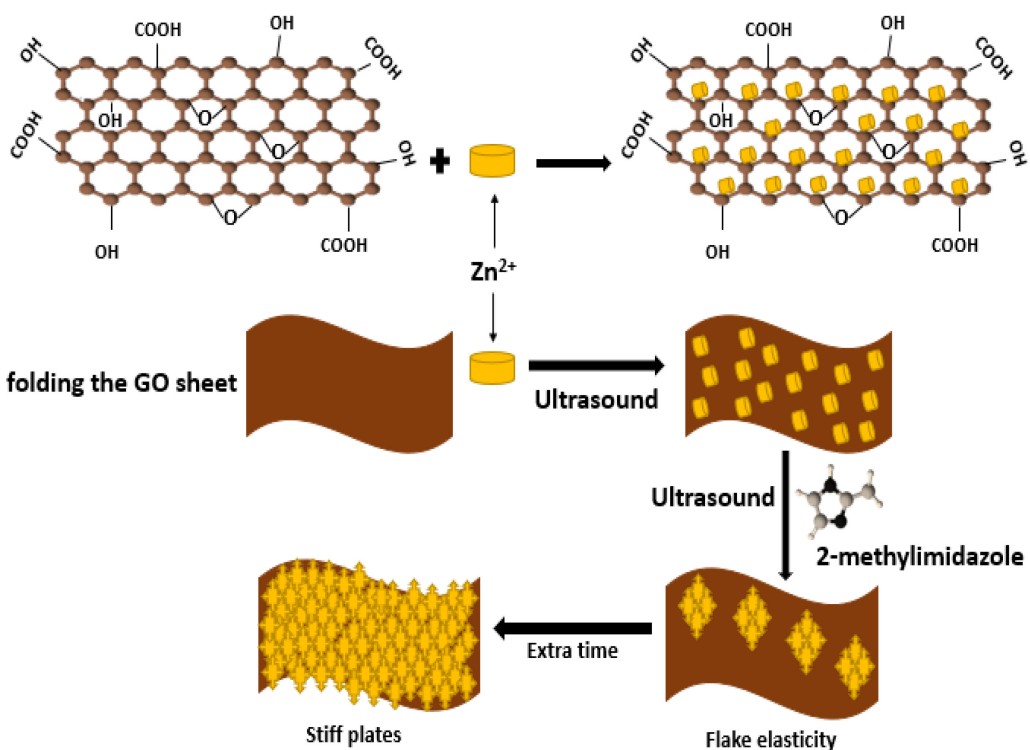

**Scheme 4.** Synthesis of GO hybrid material modified by ZIF-8.

Some of the advantages and disadvantages of the previously mentioned MOF/GO synthesis methods are briefly summarized in Table 1.

**Table 1.** Advantages and challenges of synthesis techniques for MOF/GO composites.

| Methods | Advantages | Challenges | Application | Ref. |
|---|---|---|---|---|
| Post-Synthesis | - Add new functionalities and capabilities. <br> - Improved MOFs with keeping the same structure. <br> - GO can spread between MOF crystals very well. | - Decreases the surface area. | - Gas separation and storage. | [59,60,83,84] |
| Hydrothermal or Solvothermal | - Provide a better form. <br> - Reduced agglomeration. <br> - Controlled particle form. <br> - Low energy usage. | - Poor yield. <br> - Lack of purity. <br> - Substandard in morphology and size. | - Sensors <br> - Batteries | [63,64,85,86] |
| In situ | - MOFs distributed equally on the GO surface. <br> - Strong contact forces. <br> - Eco-friendly conditions. | - Limits the growth of MOF nuclei. <br> - GO agglomeration. | - Catalysis | [68–70,87,88] |

**Table 1.** *Cont.*

| Methods | Advantages | Challenges | Application | Ref. |
|---|---|---|---|---|
| Co-Precipitation | - High purity.<br>- Simplicity of manufacture.<br>- Scalability and experimental parameter controllability.<br>- Swifter and easier. | - Requires superior solvents.<br>- Extra cleaning water and drying time. | - Batteries | [72–74] |
| Mixing | - Simple mixing.<br>- More manageable. | - Need specific conditions.<br>- Poor dispersion.<br>- MOFs and GO are not evenly matched. | - Supercapacitors<br>- Batteries<br>- Sensors | [75–77] |
| Ultrasonication | - Adjustable sizes.<br>- Quick synthesis.<br>- Simple.<br>- Eco-friendly.<br>- Mild condition.<br>- MOFs have an intact structure. | - The particle size of MOFs has drastically decreased. | - Water purification | [78–82] |

Only with the support of advanced characterization techniques such as X-ray photoelectron spectroscopy (XPS) and high-resolution transmission electron microscopes (HRTEM) were the structures of GO and rGO determined. Therefore, compared to pure GO and rGO, it is more difficult to fully characterize the MOF/GO and MOF/rGO composites. The reason is that MOFs are a bit brittle, susceptible to beam irradiation, and composed of many organic components. Familiar techniques, such as thermogravimetric analysis (TGA), infrared spectroscopy (IR), electron microscopy (EM), and powder X-ray diffraction (PXRD), typically only serve to confirm that the MOF structure has been preserved within the composite. In some instances, the PXRD pattern of the composites can still display the distinctive peak of GO or rGO, albeit slightly shifted, suggesting that there may be some variations in the interlayer distances resulting from the intercalation of MOF between the sheets. MOF/GO composites require special consideration when using TGA. It is common knowledge that GO releases $CO_2$ and CO when heated. Due to the subsequent thermal expansion, there is a substantial mass loss of about 200 °C. The material that was expelled during explosive exfoliation, however, is what caused this loss. The TGA of MOF/GO composites should not be operated at a high heating rate (above 5 C/min) under an oxygen atmosphere to deliver precise results [89,90].

## 4. Applications of Metal-Organic Frameworks/Graphene Oxide (MOFs/GO)

Graphene oxide (GO) and MOF hybrid materials, also referred to as GO/MOFs, have recently demonstrated promise in numerous applications, including supercapacitors, gas separation storage, water purification, sensors, catalysis, batteries, biomedical, and many others, as illustrated in Figure 8. Uncommonly, these hybrid composites have become viable options for environmental restoration projects that are both economical and environmentally friendly [91,92].

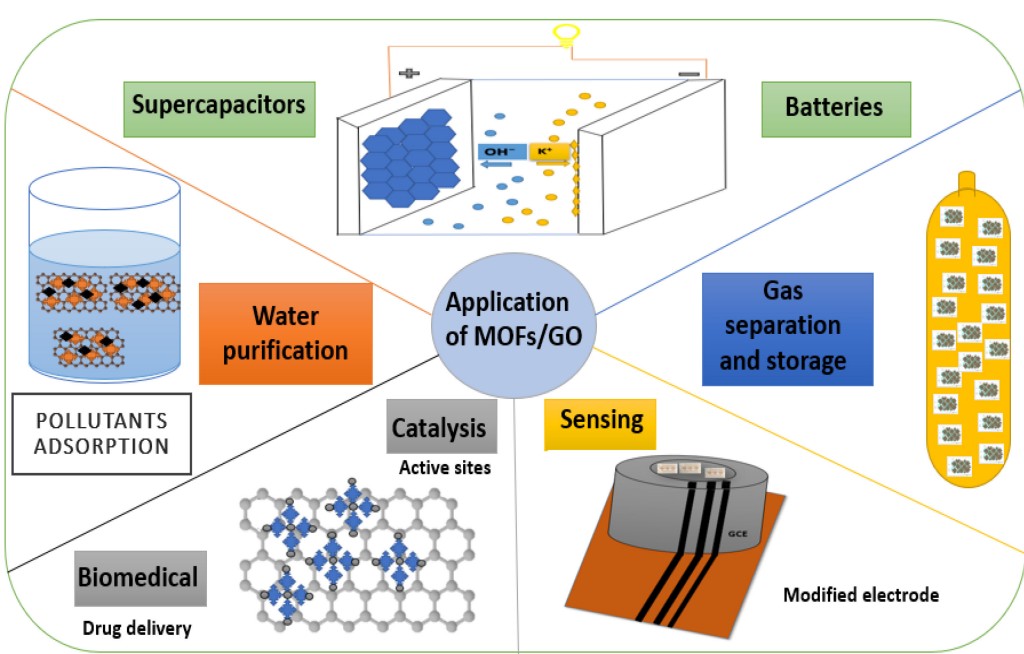

**Figure 8.** Some applications of MOFs/GO.

*4.1. Supercapacitors*

Due to their extensive power capacity, outstanding safety, and steady cyclability, supercapacitors (SCs) are electrical energy storage (EES) devices with a promising future [93]. Recently, research has demonstrated that doping heteroatoms or ions can significantly improve the electrical characteristics of MOFs in SCs. One of the most effective methods to overcome the low conductivity of unspoiled MOFs is to use various conductive carbonaceous materials, such as AC, graphene, and NTS, to manufacture electrode materials (for high-performance SCs) [94]. For many device applications, graphene's superior electrical characteristics are crucial. GO is a perfect electrode material for supercapacitors because of its remarkable mechanical flexibility, enormous specific surface area, good electron conductivity, and appropriate porosity [95,96].

Two-dimensional MOFs have been proposed as promising electrodes for a high-performance supercapacitor due to their numerous active sites, high accessible surface area, high electron transfer rates, and quick ionization. Hence, due to the low conductivities of the preceding materials, their applications are limited. Li and colleagues [97] produced 2D/2D NiCo-MOF/GO (NCMG) hybrid nanosheets via an easy ultrasonic technique. The 2D NiCo-MOF is produced by the ultrasonic treatment, which also causes the uniform dispersion of GO on its surface. GO nanosheets were evenly distributed on the uppermost NiCo-MOF surface, creating a conductive network. Thus, this efficiently inhibits the aggregation of NiCo-MOF nanosheets and improves their electronic conductivity. The findings demonstrate the NCMG-10 has an exceptional rate capability and a high specific capacity of 413.61 $Cg^{-1}$ with 0.5 $Ag^{-1}$. The assembled NCMG-10/AC asymmetric supercapacitor (ASC) provided suitable cycling stability and an extreme energy density of 36.83 $Whkg^{-1}$. When compared to pristine MOFs, MOF-based composites have a better rate capability, a long cycling life, and a large specific capacitance [97].

Chen and others [98] have synthesized extremely porous 3D structures by combining Co-BTC and GO to fulfill the demand for high-performance supercapacitors. The Co-BTC 3D microspheres and the 2D lamellar structure of GO were fused to produce composites with superior electron and charge transport capabilities. In the comparison of pure GO and Co-BTC, the Co-BTC@GO composite had a stable interface between the electrodes and electrolyte and a better charge transfer path. Co-BTC@GO 2 (0.02 g of GO) demonstrated the best energy storage performance. The highest possible exact capacitance, at 1 A/g, was 1144 F/g, and the rate capability was excellent. Co-BTC@GO 2 has a remarkably stable

life of 88.1%, even after 2000 cycles. The advantages of this electrode material are that it contains MOFs with organic linkers to control the porous structure and GO with numerous functional groups for mechanical strength [98].

Specific Features of MOF-GO as Supercapacitors

Combining MOFs with carbon materials, such as AC, CNTs, hydrogels, carbon aerogels, and graphene, has been reported to establish composites. These composites have many properties, including significantly improved electrical properties while amplifying or retaining other mechanical, chemical, thermal, and electrical characteristics. These composites have gained popularity because they preserve the proper pore size distribution, which is necessary to produce effective supercapacitor electrodes [99].

Researchers have been paying close attention to using electrode materials for supercapacitors based on MOFs because of their remarkable tunable pore size, surface area, and abundance of redox sites. Incorporating graphene nanoplatelets (GNP) into a hierarchical Ni-MOF/GO demonstrated a singular synergistic interaction between two different graphitic carbon structures in response to unstable and weakly electrically conductive properties of pristine MOFs. Along with the increasing conductivity and stability, adding GO and GNP to MOF also improves the interfacial interaction and transport kinetics for electrons and ions between supercapacitor electrodes. Ibrahim et al. (100) synthesized supercapacitors by straightforwardly fabricating a hierarchical Ni-MOF/GO/GNP electrode, mounting it in symmetrical coin and pouch cells, and using 2.0 M potassium acetate as the electrolyte. The MOF's structural stability and electrical conductivity, however, need improvement. As a result, GO with enough oxygen functional groups could form a variety of interactions, such as chemical bonding and electrostatic interaction with exposed $Ni^{2+}$ ions of Ni-MOF, which could address the problem with MOF's lack of structure stability. The highest dispersion in solvent chemical modification, exceptional chemical stability, excellent colloidal solution stability, and remarkable mechanical strength have all demonstrated that GO is the most effective structure for the fabrication of MOFs' hybrid electrodes. Furthermore, GO has excellent mechanical and optical qualities for many applications [100].

The poor stability during the charge/discharge process that MOFs and materials derived from MOFs may cause has been regarded as the major constraint in supercapacitors. Thankfully, MOFs with carbon components such as CNT, GO, and graphene could improve the electrochemical performance of supercapacitors. As a result, research has shifted to creating MOF/carbon composites. Ramachandran et al. (101) used a wet chemical process to produce Ce-MOF/GO and Ce-MOF/CNT to improve the electrochemical characteristics of supercapacitors. In both electrolytes, the Ce-MOF/GO composite showed that the electrochemical behavior was improved, and the highest specific capacitance of 2221.2 F $g^{-1}$ was obtained, with an energy density of 111.05 W h $kg^{-1}$ at a current density of 1 A $g^{-1}$ [101].

The production of advanced materials for MOFs has recently attracted interest in the fabrication of SCs due to their remarkable porosity and large surface area. Although MOFs have a high capacity and performance, their applications are limited by their poor mechanical properties, low electrical conductivity, and insufficient stability. Due to its 2D structure, superior electrical conductivity, and excellent chemical and mechanical properties, graphene (G) is most frequently used to enhance the electrochemical properties. However, extreme restacking of graphene sheets reduces the electrolyte's accessible surface area during composite processing, which lowers the capacitance. Guest materials are incorporated into the composites to address this problem. The mechanical and electrical properties of the graphene sheets will benefit from these materials' ability to prevent restacking. Azadfalaha and associates [102] used a one-step, in situ synthesis of cobalt-based MOFs with graphene (CoMG nanocomposite) to improve MOFs' lacking qualities. Due to the complementary properties of CoM and graphene, the nature of faradaic charge storage, and the mechanical stability of the composite structure, the CoMG5/CA cell exhibits an excellent electrochemical performance. Adding graphene shortens the ion

transport pathway, restricts volumetric changes during the redox reaction, and makes the composite more mechanically stable. The highest capacity of the CoMG5 electrodes was 549.96 F g$^{-1}$ F g1, and they were used as an electrode material for SCs. At a voltage window of 1.7 V, the CoMG5//CA asymmetric cell displayed a specific capacity of 50.20 F g$^{-1}$ at 20 mV s$^{-1}$, an energy density of 8.1 Wh kg$^{-1}$, and a power density of 850 W kg$^{-1}$ at 1 A g$^{-1}$. Additionally, it demonstrated a 78.85% appropriate cycling stability after 1000 charge/discharge cycles [102].

### 4.2. Gas Separation and Storage

Composite materials based on MOFs are observed to be a promising class of materials because they have the potential to combine the benefits of MOFs and other elements. Additionally, the disadvantages of each component are individually minimized in adsorption and separation systems [103]. MOFs are the dominant component of MOFs/GO composites, which produce admixtures rather than composites with covalent bonds. By using the GO substrate as a ruling component and support, nanoscale MOFs will be chemically bonded over the top of the platform [104].

MOF materials offer a lot of benefits for CO$_2$ capture. Al (HCOO)$_3$ (ALF), a form of aluminum, is also among the most basic MOFs. To address its cost and scalability, Evans et al. [105] produced ALF characteristics from readily available and affordable starting materials, such as formic acid and aluminum hydroxide, as mechanical materials to examine the adsorption [105]. To increase the capabilities of MOFs for the adsorption of CO$_2$, Zhao et al. [106] produced a composite material of L-arginine-modified MOFs with GO (MOFs/GO-Arg). The bridging of L-arginine-modified GO with the linker MOF (Cu-BTC) was chosen due to its stability, simplicity of synthesis, and affordability. By adjusting the pore shape and chemical environment at the interface between MOFs and GO, the performance of CO$_2$ adsorption can improve. Additionally, L-arginine was selected due to its biodegradability, low toxicity, and eco-friendliness. The outcomes of the MOF/GO-Arg demonstrated that the adsorption mechanism was chemical and physical co-adsorption [106].

Azizi et al. [107] developed a CuBTC/GO composite. Then, the next step was to incorporate CuBTC and CuBTC/GO composites into the polysulfone (PSF) polymers to synthesize MMMs. Using a CuBTC/GO composite as a filler in PSF resulted in an astounding increase in selectivity and gas permeability. The pre-produced CuBTC/GO composite was added to a PSF matrix using different incorporation loadings for synthesizing MMMs. The highest permeability with a 15 wt% loading of fillers in MMMs showed that the permeability of MMMs for all gases was increased compared to pure PSF [107].

### 4.3. Water Purification

To effectively remove different pollutants from polluted water and wastewater streams, MOF materials, a hybrid class of substances with a metallic center and organic linkers, are incredibly effective due to their porous nature [108]. Pharmaceuticals, particularly antibiotics, are removed from contaminated water using MOF-based adsorbents and catalysts [109]. Sewage treatment employs a variety of methods to remove environmental pollutants. Separation, absorption, photocatalytic degradation, and membrane filtration are the typical techniques [110]. The development of GO/MOFs, which have higher hydrophilicity, fouling resistance, and selectivity, has sparked interest in the water industry [111].

Textile wastewater is one of the most challenging for wastewater treatment due to the highly contaminating solid pollutants, such as dyes. In another study, Jafarian et al. [112] synthesized a novel NF membrane to remove DIRECT RED 16 (DR16) dyes and humic acid from synthetic wastewater. To form this membrane, they deposited a thin layer of GO and a Zn-based metal-organic framework (ZIF-7) on a chitosan (CTS)-coated polyethersulfone (PES) substrate. The surface membrane was made rougher and more hydrophilic by adding GO-ZIF-7. According to the data, the 5GO-ZIF membrane (~94%) had the highest dye clearance rate. Additionally, the highly hydrophilic surface of the GO-ZIF layer and the

biocidal activities of GO and zinc introduced by the GO-ZIF-7 nanocomposite improved the antifouling and anti-biofouling capabilities of the membrane. It was observed that overlaying the CTS membrane with the 5GO-ZIF nanocomposite layer lowered the pure water flux (11.4%), which may be a drawback for actual industrial applications. The increase in mass transfer resistance caused by the addition of layers comprising the GO-ZIF nanocomposite can be used to explain the decrease in pure water flux [112].

Due to their instability in aqueous solutions, GO membranes are not applicable for membrane separation technology-based wastewater treatment. Through filtering the corresponding Sm-MOF/GO dispersions, Yang et al. [113] were able to successfully create a nanocomposite membrane (Sm-MOF/GO) made of samarium metal-organic frameworks (Sm-MOFs) and GO nanosheets. The GO sheets were protected from adjacent GO layers expanding in aqueous solutions by the in situ growth of Sm-MOF with aqueous stability, which gave the prepared Sm-MOF/GO membrane a stable membrane skeleton structure. The combination of the Sm-MOF framework structure with the GO lamellar structure resulted in the M-0.31 composite having steady permeance (26 L $m^{-2}$ $h^{-1}$ $bar^{-1}$) and rejection (>91%) to the organic dye rhodamine B (RhB) after the optimization of the Sm-MOF loading contents. The exclusion remained mostly confined without significant decline after 5.5 h of continuous filtration, demonstrating the long-term stability of M-0.31. Thus, the fake M-0.31 might be a candidate for specific wastewater treatment [113].

*4.4. Sensors*

MOFs are combined with some conducting materials, such as carbon and conducting polymers, to produce materials for electrochemical sensors with good analytical performance. Graphene is an example of a well-known 2D substance that has gained interest due to its excellent electrical conductivity and surface area. By contrast, MOFs and graphene produced new physical and chemical properties [114]. Due to their substantial specific surface area, high electrical conductivity, and superior chemical stability, graphene and its analogs are widely employed to develop electrochemical sensors. The MOF conductivity effectively increased by combining graphene and its analogs with MOFs [115].

Environmental monitoring would greatly benefit from constructing a sensitive voltametric platform to examine the excessively toxic organic pollutant p-chloronitrobenzene (p-CNB). A composite made of nickel-based MOF (Ni-MOF) and GO was successfully synthesized by Gao et al. [116] using simple and affordable chemical precipitation. The Ni-MOF/GO composite was modified to produce an electrochemical sensor for p-CNB. With the help of electrochemical impedance spectroscopy, distinguishing pulse voltammetry and cyclic voltammetry were used to examine the electrochemical characteristics of as-prepared sensors. Thus, the detection limit of the composite sensor was 8.0 nM (S/N = 3), and it demonstrated superior electrocatalytic performance towards the oxidation of p-CNB in the concentration range of 0.10–300.0 µM. Subsequent research revealed the production sensing platform to have good long-lasting stability, strong selectivity, and reproducibility [116].

A sensitive electrochemical sensor was produced to quickly determine the presence of bisphenol A (BPA) based on a cerium-centered, metal-organic framework, electrochemically reduced graphene oxide composite (Ce-MOF-ERGO) and cetyltrimethylammonium bromide (CTAB) signal amplification. Wang et al. [117] fabricated the Ce-MOF-ERGO composite by electrochemically reducing the Ce-MOF-GO composite. The combination Ce-MOF-ERGO composite offered enough electrocatalytic activity and reactive sites for BPA. Furthermore, adding CTAB to the electrolyte solution significantly improved the BPA oxidation signals on Ce-MOF-ERGO-modified electrodes. The sensor demonstrated high sensitivity for BPA detection and was successfully applied to detect BPA in fundamental samples with good recoveries based on the dual-signal-amplification strategy [117].

*4.5. Catalysis*

In many diverse industries, particularly the chemical sector, catalysts (homogeneous and heterogeneous) play a significant role. The industrial use of homogeneous catalysts is

restricted because they frequently have a higher catalytic activity and are challenging to recover from the reaction solution. Contrarily, heterogeneous catalysts (supported catalysts) are easily removed from the reaction solution and repeatedly used. These catalysts have received significant attention when used in different reaction systems [118,119]. Catalytic processes are necessary for over 90% of chemical industrial processes and over 20% of all chemical products. Catalyst production is a crucial process that requires extensive research to produce the high-performance catalysts currently in use [120]. The MOFs-GO have become a new class of catalytic composite materials in the nanoarchitecture field because of their high surface area, unique electrical properties, magnificent conductivity, and hydrophilic nature. Nano-structuring porous catalysts would be a crucial way to expose accessible active sites to achieve high catalytic activity [121]. MOFs-GO is a hybrid material that combines the distinct benefits of MOFs and GO and is ideal for immobilizing nanoparticles (NPs) [122]. Additionally, this pairing of MOFs and GO provides a practical path to heterogeneous catalysts with high performance and stability [123]. Additionally, because of their porous structure, MOFs-GO materials have excellent applications in electrocatalysis and photocatalysis [124].

### 4.5.1. Electrocatalysts

The study of environmentally friendly energy has received significant attention from researchers due to an evident trend of rising energy consumption. The hydrogen evolution reaction (HER), the oxygen evolution reaction (OER), and the oxygen reduction reaction (ORR) are consequential electrochemical reactions used in fuel cells and batteries for energy storage. Graphene-based materials are frequently used in composite MOFs to create efficient electrocatalysts because of their low cost and high carrier conductivity [125–127].

Gopi and colleagues [128] produced the catalyst V-Ni $_{0.06}$ Fe$_{0.06}$ MOF/GO using an in situ technique. They performed excellent bifunctional electrocatalysis for the hydrogen evolution reaction (HER) and the anodic oxygen evolution reaction (OER) and showed high durability in both acidic and alkaline mediums. The choice of V, Ni, and Fe redox metal nodes in combination with a highly porous MOF/GO composite for the water splitting reaction is motivated by the abundance of free carriers at the Fermi level of the V atom and the more active edge sites of the Fe atom. Moreover, the MOF-to-graphene MOF conducting mechanism was improved by 2D-graphene-combined MOF composites. Additionally, it is cheap, easy to make, has a large surface area, and primarily possesses magnetic properties. The BJH pore distribution plot and BET N2 adsorption–desorption isotherms were used to determine the V-Ni$_{0.06}$ Fe$_{0.06}$ MOF/GO catalyst's BET surface area. The BET surface area of the V-Ni$_{0.06}$ Fe$_{0.06}$ MOF/GO sample is 21.7 m$^2$ g$^{-1}$. For efficient seawater electrolysis, a very promising electrocatalyst has been proposed. Due to its straightforward nature, high activity, corrosion stability, and scalable synthesis process, a non-noble NiFe metal-based bifunctional water-splitting catalyst is a promising electrocatalyst for hydrogen production [128].

Noor et al. [129] used the improved Hummers method to prepare GO and a simple hydrothermal method to assemble Cu-MOF/GO composites. In the methanol oxidation reaction, electrocatalysts were constructed using a copper benzene tricarboxylic acid MOF (Cu-BTC MOF). The composites exhibited synergistic properties due to the high surface area and exceptional conductance of MOFs and GO, respectively. The highly porous, 3D MOFs demonstrated phenomenal electrocatalytic performance in the methanol oxidation reaction. Moreover, to make the direct methanol fuel cell (DMFC) commercially viable, the GO/Cu-MOF exhibited remarkable potential as a substitute for the current Pt-based, expensive, and rare catalysts. The highest current density of 120 mA/cm$^2$, at a scan rate of 50 mV/s and a voltage of 0.9 V, is shown by the 5 wt% GO/Cu-MOF catalyst series, out of all the catalyst series [129].

4.5.2. Photocatalysts

According to researchers, MOFs are highly efficient in the photocatalytic degradation of pollutants and organic dyes when used as catalysts in photocatalytic processes. The retrieval and separation of photocatalyst MOFs from the reaction mixture can be facilitated and improved through modifications. Thus, the performance of photocatalysis can be enhanced by GO [130,131].

To improve the catalytic performance of MOFs, Jin et al. [132] intensively hybridized them with GO; however, it is still unclear how the pore structure of MOFs affects the activity of GO/MOF hybrids. To investigate the connection between the pore structure of MIL-125 and the activity of the hybrid, they created a variety of GO/MIL-125 hybrids using the sonication process. The obtained GO/MIL-125(H) has an electron mediator, a significant surface area, displays an assistant photothermal effect, and has hierarchical pores that aid in the adsorption and photodegradation of toluene. Additionally, they discovered that during the photocatalysis procedure, 10% GO/MIL-125 (H) had a higher surface temperature than 10% GO/MIL-125 (M), which had the effect of speeding up the charge transfer and gas diffusion. In hierarchically porous hybrids, these factors improve the photocatalytic performance [132].

The photocatalytic activities of UiO-66 and GO (UiO-66_GO) nanocomposites for the degradation of carbamazepine (CBZ) were examined in a different study by Heu et al. [133] using a one-step hydrothermal method. Under a range of circumstances, including GO loading, catalyst dosage, initial pollutant concentration, and solution pH, the nanocomposite for the degradation of CBZ was investigated. Increased surface area and porosity, a smaller bandgap, and better light absorption are all factors that were found to be responsible for GO's enhanced photocatalytic activity. Additionally, the composite demonstrated notable stability and recyclability over five successive cycles of photocatalytic degradation. Modifying semiconductors with GO as an electron acceptor is a successful method for enhancing photocatalytic activity. Additionally, it serves as a great source of inspiration for the growth of additional GO-based composite photocatalysts and the use of UiO-66 in water or wastewater treatment methods. The UiO-66_GO nanocomposites demonstrated a high overall removal efficiency (>90%) in 2 h and a photocatalytic rate constant of up to 0.0136 $min^{-1}$. The results of the experiments supported the idea of a photocatalytic mechanism for enhanced CBZ photodegradation by showing that $O_2^\bullet$ and $OH^\bullet$ are the responsible radicals for photocatalytic degradation. Further, using this composite material as a solar-base catalyst is possible due to its ability to absorb light in the visible light spectrum (400–700 nm) [133].

*4.6. Batteries*

Fuel cells and large-capacity, inexpensive, and sustainable rechargeable batteries have drawn a lot of attention due to the new energy industry's rapid development [134]. Batteries are known to store relatively large amounts of energy compared to supercapacitors but have relatively low power delivery or uptake, a short life cycle, and thermal management issues [135]. Due to their unique properties of the adaptable structure, high porosity, and abundance of active sites, MOFs have received extensive research as a typical inorganic–organic nanomaterial in the growth of battery electrodes [136]. Several studies have shown that combining MOFs with graphene-based conductive materials can significantly improve the electronic conductivity, stability of the system, and capacity [137]. The design and synthesis of 3D GO/MOF composites and their derivatives, which have excellent physical and chemical properties for electrode materials for lithium-ion batteries (LIBs), sodium-ion batteries (SIBs), potassium-ion batteries (PIBs), and lithium-sulfur batteries (LSBs), are currently the subject of a large body of research [138]. In situ growth and ex situ growth are two methods that facilitate the synthesis of MOF-graphene composite materials, which are widely used in electrochemical energy storage because they exhibit good electrochemical properties [139].

Li et al. [140] successfully produced Ni(OH)$_2$-GO electrode materials using a two-step synthesis process. The numerous functional groups on GO nanosheets aid in the nucleation and growth of Ni-MOFs. The subsequent hydrolysis of Ni-MOF makes it possible to successfully prepare Ni(OH)$_2$-GO for its promising use in supercapacitors. Within Ni(OH)$_2$-GO, the GO content decreases to 7.41 wt%. The electron transfer between Ni(OH)$_2$ and GO readily occurs during the electrochemical reaction, drastically increasing the electrochemical activity of Ni(OH)$_2$. Additionally, Ni(OH)$_2$, anchored on the GO nanosheet, prevents the 2D GO sheets from aggregating, and the preserved layered structure offers open pathways for electrolyte ions to move through and extends the cycling life. Therefore, GO can significantly increase the SC of Ni(OH)$_2$ by almost 50%, despite the slight decrease to only 7.41 wt%. The Ni(OH)$_2$-GO electrode material exhibits exceptional stability (the capacity retention rate after 8000 cycles is 108%) and a high SC of 1007.5 C g$^{-1}$. The ASC device can operate with excellent stability at a power density of 393.29 W kg$^{-1}$ and has an energy density of 65.22 Wh kg$^{-1}$ when PPD-rGO is used as the negative electrode [140].

In a different study, Wu and colleagues [141] successfully synthesized a 3D hierarchical MOF-on-rGO compartment through an in situ reduced and combined process. Its unique characteristics, which combine the polarity and porous attributes of MOFs with the high conductivity of rGO, make the MOF-on-rGO compartment a prime candidate as a sulfur host in lithium-sulfur (Li-S) batteries. The MOF-on-rGO-based electrode could accomplish a high initial discharge capacity of 1250 mAhg$^{-1}$ at a current density of 0.1 C (1.0 C = 1675 mAhg$^{-1}$). To maintain a stable cycling performance, both the polar pore environment of MOF and the hierarchical structures of rGO prevent the diffusion and migration of soluble polysulfide. The spongy-layered rGO can also buffer volume changes due to expansion and contraction, providing stable facilities for Li-S batteries [141].

### *4.7. Other Applications (Biomedical)*

Numerous MOFs and MOF-derived nanomaterials have been developed and used in biomedicine for antibacterial mechanisms [142]. In different studies, MOF composites have also been examined for bio-applications such as drug delivery, bio-imaging, and cancer treatment [143]. There is not a thorough review report available that explains how to functionalize GO with MOFs for biomedical applications or how to use natural chemotherapeutic agents for cancer therapeutics [144]. The amount of GO's active surface available for interacting with bioactive molecules was constrained by its intense agglomeration. Combining it with other NPs, such as MOFs, is an appealing strategy for overcoming this restriction and enhancing GO's efficiency in the biomedical sector [145].

To compare their effectiveness as a vehicle for the anticancer drug carriers 5-Fu, Pooresmaeil and colleagues [146] created three different types of chitosan-based microspheres: CS, chitosan-coated zinc-based MOF (CS/Zn-MOF), and a chitosan-coated hybrid of ZnMOF with GO (CS/Zn-MOF@GO). The ternary of hybrid CS/Zn-MOF@GO microspheres was found to have the highest amount of 5-fluorouracil (5-Fu) loading, at about 45%. The rough-surfaced, 5-Fu-loaded CS microspheres (5-Fu@CS/Zn-MOF@GO microspheres) displayed a pH-sensitive and sustained release pattern for the 5-Fu that was loaded. Therefore, the total amount of drug released over time at pH 5.0 was roughly twice that at pH 7.4. Ultimately, CS/Zn-MOF@GO microspheres demonstrated acceptable enzymatic biodegradability and good biocompatibility with the epithelial human breast cancer cell line MDA-MB 231. The ability of 5-Fu-loaded CS/Zn-MOF@GO microspheres to treat tumor cells was demonstrated by the cell viability of 41.2% following 48 treatments with 5-Fu@CS/Zn-MOF@GO microspheres [146].

Another study found that the development of antibacterial dressings has become a tactic to control wound infections brought on by bacteria resulting from the spread of drug-resistant bacteria in hospitals. Zhang et al. [147] successfully assembled a sericin/chitosan/Ag@MOF-GO (CS/SS/Ag@MOF-GO) nanocomposite using an easy, one-step procedure. Investigations were conducted to determine how the presence of GO and Ag@MOF affected the samples' microstructure, swelling level, biocompatibility, water-solubility, hemostatic

activity, water retention, antibacterial activity, cell adhesion, cell migration, and results from animal experiments. The composite had a 3.9% hemolysis rate and a 131.2% cell viability rate, which showed the hydrogel to have good biocompatibility. An in vitro cell migration assay and a mouse in vivo evaluation suggested and confirmed that the composite could hasten wound healing and re-epithelialization. In conclusion, the composite material makes a fantastic dressing for promoting wound healing, preventing bacterial infection, and speeding up hemostasis [147].

Table 2 summarizes MOFs-GO applications, including composite materials the output, and synthesis methods.

**Table 2.** Summaries of some applications for MOF/GO composites.

| Applications | Composites | Output | Synthesis Methods | Ref. |
|---|---|---|---|---|
| Supercapacitors | - 2D/2D NiCo-MOF/GO | - The composite has an exceptional rate capability and a high specific capacity of 413.61 Cg$^{-1}$ with 0.5 Ag$^{-1}$. | - Ultrasonic | [97] |
| Gas separation and Storage | - CuBTC/GO | - The highest permeability with a 15 wt% loading of fillers in MMMs showed that the permeability of MMMs for all gases was increased compared to pure PSF. | - Hydrothermal | [107] |
| Water Purification | - Sm-MOF/GO | - M-0.31 composite having steady permeance (26 L m$^{-2}$ h$^{-1}$ bar$^{-1}$) and rejection (>91%) to the organic dye rhodamine B (RhB) after the optimization of the Sm-MOF loading contents. | - In situ growth | [113] |
| Sensor | - Ni-MOF/GO | - The detection limit of the composite sensor was 8.0 nM (S/N = 3), and it demonstrated superior electrocatalytic performance towards the oxidation of p-CNB in the concentration range of 0.10–300.0 μM. | - Chemical precipitation | [116] |
| Electrocatalysts | - Cu-MOF/GO | - The highest current density of 120 mA/cm$^2$, at a scan rate of 50 mV/s and a voltage of 0.9 V, is shown by the 5 wt% GO/Cu-MOF catalyst. | - Hydrothermal | [129] |
| Photocatalysts | - UiO-66_GO | - The UiO-66_GO demonstrated a high overall removal efficiency (>90%) in 2 h and a photocatalytic rate constant of up to 0.0136 min$^{-1}$. | - Hydrothermal | [133] |
| Batteries | - MOF-on-rGO | - The MOF-on-rGO-based electrode is a prime candidate as a sulfur host in lithium-sulfur, yielding a high initial discharge capacity of 1250 mAhg$^{-1}$ at a current density of 0.1 C (1.0 C = 1675 mAhg$^{-1}$). | - In situ | [140] |
| Biomedical | - CS/Zn-MOF@GO | - The cell viability of 41.2% after 48 treatments with 5-Fu@CS/Zn-MOF@GO microspheres proved the ability of 5-Fu-loaded CS/Zn-MOF@GO microspheres to treat tumor cells. | - Ultrasonic | [145] |

## 5. The Advantages and Challenges of Metal-Organic Frameworks/Graphene Oxide (MOFs/GO)

There are many applications for MOFs and materials based on graphene, especially in electrochemistry. The presence of GO during the production of MOFs has several

benefits, which include the capacity to tune the particle size, quicker electron mobility, and chemical and thermal stability. The best performance under specific environmental conditions (pH, temperature, humidity, etc.) is necessary for MOF and graphene-based composites to operate in harsh environments, thus constraining their ability to develop. In this regard, MOF/graphene-based materials still have room for improvement [148]. Additionally, the combination of GO and MOFs results in the creation of tiny holes, which leads to an increase in the dispersive strength of the MOFs, suppressed MOF aggregation, strong specific adsorption, and a high rate of $CO_2$ storage [149]. GO can resolve the weak coordination bonds between metal nodes and organic ligands, guide MOF development, and lessen poor conductivity [150]. During the production of MOF derivatives, the merge of graphene-based materials can stop high-temperature etching, break down the MOF structure, and perform other processes that would otherwise lower the specific surface area and active sites of MOFs [151]. The composite MOFs/GO have hierarchical pore structures that provide an ideal space for oxygen atoms in GO, enhancing its stability [152]. GO combined with MOFs decreases toxicity and exhibits exceptional electrochemical, mechanical, thermal, and electrical properties. Due to the diversity of MOF compounds and complex MOF–GO interactions, MOFs' growth and structure orientations are still challenging to predict [153]. Otherwise, MOF/GO composites can be more robust and stable because of the abundant organic materials on the surface of GO nanosheets, which increase the brittleness and rate of MOFs' degradation [154]. Furthermore, multifunctional MOFs can be utilized to produce multivariate materials by introducing several functional sites onto porous surfaces [155]. Besides, the functionalization of GO nanosheets' surface results in a better ability under demanding circumstances [156]. Finally, a practical and effective method to lessen GO's toxicity in natural aquatic systems is provided by MOFs-GO [157].

## 6. Conclusion and Future Perspectives

The chemical and physical properties of composites made from MOFs and GO are enhanced, enabling their use in many technological applications. The produced MOF/GO composites improve the application of MOFs by eliminating constraints associated with the separate substances (MOFs and GO). This work covered the most popular and documented synthesis techniques, including post-synthesis, in situ, hydrothermal, ultrasonication, mixing, and co-precipitation, and their applications in supercapacitors, gas storage, water purification, sensing, catalysis, batteries, and biomedical fields. In most applications, MOFs/GO are applied as adsorbent materials because they contain GO and have stronger interactions. Moreover, the advantages, challenges, and properties of MOF/GO composites were discussed. To date, the synthesis and practical applications of MOF/GO composites continue to be hindered by several difficulties and challenges. The speed and scope of MOFs' development is difficult to monitor. Therefore, there are still more challenges for MOFs/GO or MOFs/rGO, including the difficulty of fully characterizing these composites compared to pure GO and rGO because MOFs are more delicate, sensitive, and fabricated by numerous organic components. Furthermore, the accumulation of GO due to large-scale directional interactions, which restrict the number of active sites and reduce the specific surface area, poses a significant obstacle to the use of GO.

In the future, the selectivity, repeatability, practicability, and stability of composite materials can all improve to increase the level of industrial application for these high-performance materials. The MOF/GO composite materials field will continue to observe rapid development because of its fascinating applications. Numerous MOF types could afford composites more potential than the few types used in recent studies. In many industrially significant applications, such as electrochemistry, photocatalysis, and gas separation and storage, the ability to synthesize MOF/GO structures and properties to meet particular requirements may become possible. The final MOF/GO composites' design and MOF selection can help customize the properties. From this study, of all the synthesis techniques mentioned above, post-synthesis may be the most suitable for industrial applications due

to the easy control of the number of MOFs and GO. With such control over the synthetic composites, it is possible to produce materials with a sizable surface area and many active sites that speed up the reaction rate in different chemical reactions. Therefore, this technique can be used to prepare the composite to meet the demands of a particular application. Finally, extensive research is still required. There are reasons to think that this technology will advance in upcoming studies, and there is a dire need for more thorough research in this area.

**Author Contributions:** Conceptualization S.A.E.N., K.O.B. and L.K.; validation S.A.E.N., K.O.B. and L.K.; investigation, S.A.E.N.; writing original draft preparation, S.A.E.N.; writing-review and editing, K.O.B. and L.K.; visualization, S.A.E.N.; supervision, K.O.B. and L.K.; funding acquisition, L.K. All authors have read and agreed to the published version of the manuscript.

**Funding:** This work was financially supported by the National Research Foundation (NRF), South Africa, grant number: 138079, and the Tertiary Education Support Program (TESP), Eskom Holdings SOC Limited, Reg No. 2002/015527/06.

**Institutional Review Board Statement:** Not applicable.

**Informed Consent Statement:** Not applicable.

**Data Availability Statement:** Not applicable.

**Conflicts of Interest:** The authors declare no conflict of interest.

## Glossary

| | |
|---|---|
| Metal-organic frameworks | MOFs |
| Graphene oxide | GO |
| Carbon nanotubes | CNTs |
| Activated carbon | AC |
| Unsaturated metal sites | CUSs |
| Nanofiltration | NF |
| Solid-phase microextraction | SPME |
| Gas chromatography | GC |
| Reduced graphene oxide | rGO |
| Nonsteroidal anti-inflammatory drugs | NSAIDs |
| Post-synthetic modification | PSM |
| Oxygen evolution reaction | OER |
| Catalytic wet peroxide oxidation | CWPO |
| Zeolitic imidazolate frameworks | ZIFs |
| Polyether sulfone | PES |
| Ultrafiltration | UF |
| Triethylamine | TEA |
| Zirconium-porphyrin MOF | PCN-222 |
| 1,3,5-Benzenetricarboxylic acid | BTC |
| Epoxy resin | EP |
| Limited oxygen index | LOI |
| Peak heat release rate | pHRR |
| Total heat release | THR |
| Maximum value of smoke density | Ds,max |
| Glassy carbon electrode | GCE |
| Mixed-matrix membranes | MMMs |
| Ethyl cellulose | EC |
| Supercapacitors | SCs |
| Electrical energy storage | EES |
| Asymmetric supercapacitor | ASC |
| Graphene nanoplatelets | GNP |
| Graphene | G |
| Polysulfone | PSF |

| | |
|---|---|
| DIRECT RED 16 | DR16 |
| Chitosan | CTS |
| Rhodamine B | RhB |
| Polyethersulfone | PES |
| p-Chloronitrobenzene | p-CNB |
| Cetyltrimethylammonium bromide | CTAB |
| Bisphenol A | BPA |
| Hydrogen evolution reaction | HER |
| Oxygen reduction reaction | ORR |
| Direct methanol fuel cell | DMFC |
| Carbamazepine | CBZ |
| Lithium-ion batteries | LIBs |
| Sodium-ion batteries | SIBs |
| Potassium-ion batteries | PIBs |
| Lithium-sulfur batteries | LSBs |
| Nanoparticles | NPs |
| Persulfate | PS |
| Trichlorophenol | TCP |
| Peroxymonosulfate | PMS |
| Tetracycline | TC |
| Density functional theory | DFT |

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
