# Peer review of "Synthesis, Properties, and Applications of Metal Organic Frameworks Supported on Graphene Oxide"

_coatings, doi:10.3390/coatings13081456_

Round 1
Reviewer 1 Report
The manuscript titled "Synthesis, properties, and applications of metal-organic frameworks supported on graphene oxide" focuses on the fascinating topic of MOF-GO composite materials. It discusses various synthesis methods and their applications in supercapacitors, gas storage, pollution removal, sensing, and catalysis. However, the review needs a clearer direction and recent advancements in the field. The authors have primarily focused on synthesis rather than applications, but the properties are more critical. I strongly recommend revising the manuscript to include a thorough examination of GO-MOF applications. After careful revision, I recommend addressing the following major points:
- Authors are recommended to provide a history and background of GO-MOFs in the manuscript to give more visibility for the review and a comprehensive focus for the readers.
- As the introduction to the MOFs area, Figure-1 should be more detailed. Please refer to some recent papers for guidance. Figure-1 has poor quality and needs to be revised or provided with a new image. Also, revise Figure-2 and ensure the images are clearly illustrated, accompanied by explanatory text and captions.
- Authors are recommended to compare the performance of early and recent pristine MOFs, such as SIFSIX and ALF. Present a comprehensive comparative table in the review. Also, authors are recommended to cite the below recent state-of-the-art MOFs that align with this topic. (Sci. Adv. 8, eade1473 (2022), J. Am. Chem. Soc. 2023, 145, 11643−11649, and J. Am. Chem. Soc. 2023, 145, 9850−9856).
- Authors should present a literature comparison table for different synthesis methods and thoroughly discuss the factors that enhance properties such as supercapacity, gas storage, pollution removal, sensing, and catalysis.
- Authors need to correct grammar mistakes and remove redundant paragraphs in the manuscript.
- Ensure the text and figures align correctly, as there are significant deviations in the current manuscript—present full figures, as they provide a clear understanding of the text. Based on the present manuscript, it is challenging to comprehend the information.
- The text in Figure 4 is difficult to read. Authors are strongly recommended to revise figure-4 and all the necessary figures where they are not clearly presented.
- Recommended to present a thorough discussion on the basic and advanced techniques used and required to analyze composite MOF-GO materials; these need to be presented in the manuscript.
- Present in a separate paragraph the reasons why the MOF-GO area faces hindrances despite its excellent chemical and physical properties
- Although this review briefly covers MOF and GO composite materials, it needs to mention future directions more specifically. Revisit the manuscript and explicitly mention the future of GO material for energy applications. Authors are strongly advised to revise the conclusion of the manuscript, which currently resembles an introduction. Provide detailed future directions along with concise achievements in the field of MOF-GO composite materials.
Authors need to correct grammar mistakes and remove redundant paragraphs in the manuscript.
Author Response
Please finds the attachment.

Reviewer 2 Report
The manuscript is a review on “Synthesis, properties, and applications of metal organic frameworks supported on graphene-oxide”.
Reviewer comments
The review - manuscript is based on 115 papers published between 2018 and 2023, inclusive. As the authors mentioned in the abstract is a review paper on recent research on MOFs/GO which is of importance for the scientific community working in this field.
However, several important aspects need to be improved.
1. The authors review on several production methods and applications. However, it is not clear which method should be used for a specific application. In table 1, it would be important to insert a column regarding the suitable applications related with a specific method of production.
2. In the conclusion it would be convenient to have a final statement referring, which of all methods described, would be suitable or more appropriate for production and synthesis at industrial level and for which industrial applications.
3. There are too many subjective and vague sentences, which should be changed.
· pg.2 line 78: “great specific surface area”- what is considered great? Please give an order of magnitude.
· Pg. 2 line 88: “…and their origin is crucial,” what the authors mean with origin in this context? Is it the process? Is it the kind of catalytic elements?
· Pg. 3, line 144 “…excellent thermal and electrical conductivity…” which are the orders of magnitude?
· Another example Pg.4 lines 162 – 164”: Graphene has been demonstrated to possess several desirable qualities, including: great mechanical strength, electrical conductivity, the capacity to form molecular barriers, and other outstanding characteristics [37]”. What are great mechanical properties and other outstanding characteristics?
More examples can be found along the manuscript.
Recommendation to review these types of sentences and, whenever is possible, to add orders of magnitude and compare the results among the works, regarding the properties, the advantages, the costs of the production.
4. Repetition of the same reference sequentially and in the same paragraph. There are entire paragraphs based on the same reference and the references are repeated several times.
· Pg. 2, lines 120 – 127: the same reference and appears 4 times.
“Hence, there are several restrictions on specific approaches to the synthesis of MOFs employing soluble metal precursors [23]. The dissolved metal ions react with the bridging linkers at an accelerating pace, leading to side reactions and the consequential production of materials that are often low in purity and high in disorder [23]. Solvent recovery may become complicated when ionic metal salts undergo rapid dissolution, leading to diffusion and oxidation issues [23]. It is very difficult to control MOFs crystal development during synthesis because conventional soluble metal precursors are challenging to work with and exploit in both space and time [23].”
· Reference [55] appears 5 times in 7 text lines. All the paragraph is based on the same reference.
· Same for the references [56], [57], [63], …
5. Fig. 2 is not necessary, since is repeated in Fig.3

No comments regarding the quality of the English.
Author Response
- The authors review on several production methods and applications. However, it is not clear which method should be used for a specific application. In table 1, it would be important to insert a column regarding the suitable applications related with a specific method of production.
➢ The table has been updated as advised
- In the conclusion it would be convenient to have a final statement referring, which of all methods described, would be suitable or more appropriate for production and synthesis at industrial level and for which industrial applications.
➢ The final statement has been updated as advised
- There are too many subjective and vague sentences, which should be changed.
- pg.2 line 78: “great specific surface area”- what is considered great? Please give an order of magnitude.
- Pg. 2 line 88: “…and their origin is crucial,” what the authors mean with origin in this context? Is it the process? Is it the kind of catalytic elements?
- Pg. 3, line 144 “…excellent thermal and electrical conductivity…” which are the orders of magnitude?
- Another example Pg.4 lines 162 – 164”: Graphene has been demonstrated to possess several desirable qualities, including: greatmechanical strength, electrical conductivity, the capacity to form molecular barriers, and other outstanding characteristics [37]”. What are great mechanical properties and other outstanding characteristics?
More examples can be found along the manuscript.
Recommendation to review these types of sentences and, whenever is possible, to add orders of magnitude and compare the results among the works, regarding the properties, the advantages, the costs of the production.
➢ All subjective and vague sentences have been updated as advised
- Repetition of the same reference sequentially and in the same paragraph. There are entire paragraphs based on the same reference and the references are repeated several times.
- Pg. 2, lines 120 – 127: the same reference and appears 4 times.
“Hence, there are several restrictions on specific approaches to the synthesis of MOFs employing soluble metal precursors [23]. The dissolved metal ions react with the bridging linkers at an accelerating pace, leading to side reactions and the consequential production of materials that are often low in purity and high in disorder [23]. Solvent recovery may become complicated when ionic metal salts undergo rapid dissolution, leading to diffusion and oxidation issues [23]. It is very difficult to control MOFs crystal development during synthesis because conventional soluble metal precursors are challenging to work with and exploit in both space and time [23].”
- Reference [55] appears 5 times in 7 text lines. All the paragraph is based on the same reference.
- Same for the references [56], [57], [63], …
➢ All references have been updated as advised
- Fig. 2 is not necessary, since is repeated in Fig.3
➢ Figure 3 has been updated as advised
Please finds the attachment.

Reviewer 3 Report
In the article, the authors reviewed the properties, advantages, and applications Metal-Organic Frameworks/Graphene Oxide (MOFs/GO). I found some problems in section 5, which is briefly discussed. I recommend the acceptance of this manuscript in Coatings after addressing the following comments.
1. In the abstract, it has been mentioned that “In this review paper, an effort has been made to outline recent research on MOFs/GO materials features, advantages, properties, synthesis techniques, and applications in a variety of industries……”
However, you have only reviewed MOFs/GO for several applications, not for industrial applications. Make it clear and correct it if required.
2. In this review, the progress of MOFs/GO has been carried out for several applications such as supercapacitors and so on. Nowadays, MOFs have received significant attention for several Biomedical applications such as disease diagnosis and drug delivery. I recommend the addition of the biological application of MOFs to section 5.
3. In section 5 the applications of Metal-Organic Frameworks/Graphene Oxide (MOFs/GO) have been discussed. I recommend the addition of a new table that summarizes section 5 including materials, their applications, the results/output, reference, etc. Such a table will be helpful for the readers to understand the review more conveniently.
4. I couldn’t see the reprint statement for the figures in their caption. If the figures are taken from any other articles, it is necessary to make a reprint statement and cite the article.
5. Section ‘5.5 Catalysis’ is too briefly discussed. MOFs are quite famous materials for several catalytic applications including H2 evolution, O2 evolution, CO2 reduction, N2 reduction, etc., However, in this section, only organic compound degradation has been discussed. I strongly recommend shedding light on as mentioned catalytic applications.
6. Moreover, there are several branches of catalysis, such as photocatalysis, electrocatalysis, thermocatalysis, biocatalysis, etc. which are necessary to differentiate.
7. Section 5.3 Pollutant removal is also a part of Catalysis, then why you separately discussed it? If you see carefully, in the catalysis section also, you discussed pollutant degradation. For example, “As a result, the corresponding TC removal and mineralization efficiencies increased to 91.66% and 45.04%, respectively.”
8. A couple of sentences are hard to understand, make them accessible and understandable to the readers. For example, in the sentence “GO has drawn significant research interest due to its surface area resembling MOFs and the particular processing and handling advantages of dispersion and amphiphilicity in water provided by the oxygen-containing groups present” there are many ‘and’.
9. There are several typos in the manuscript, correct them.
For example, (a) in the sentence ‘D’ of due should be a small case in the following sentence, “MOF-based GO has recently attracted attention Due to its resemblance to metal ions and organic binding linkers.
The quality of English is good, but some minor typos need to be corrected.
Author Response
Please finds the attachment.
- In the abstract, it has been mentioned that “In this review paper, an effort has been made to outline recent research on MOFs/GO materials features, advantages, properties, synthesis techniques, and applications in a variety of industries……”
➢ Has been updated as advised
However, you have only reviewed MOFs/GO for several applications, not for industrial applications. Make it clear and correct it if required.
➢ Has been updated as advised
- In this review, the progress of MOFs/GO has been carried out for several applications such as supercapacitors and so on. Nowadays, MOFs have received significant attention for several Biomedical applications such as disease diagnosis and drug delivery. I recommend the addition of the biological application of MOFs to section 5.
➢ Has been updated as advised
- In section 5 the applications of Metal-Organic Frameworks/Graphene Oxide (MOFs/GO) have been discussed. I recommend the addition of a new table that summarizes section 5 including materials, their applications, the results/output, reference, etc. Such a table will be helpful for the readers to understand the review more conveniently.
➢ Has been updated as advised
- I couldn’t see the reprint statement for the figures in their caption. If the figures are taken from any other articles, it is necessary to make a reprint statement and cite the article.
Regarding all the figures and schemes in my review paper, I made them myself and used the references to illustrate the process of the methods, except scheme 1, which has copyrights I received the permission through an email due to it not being available on RightsLink to obtain printable license permission.
- Section ‘5.5 Catalysis’ is too briefly discussed. MOFs are quite famous materials for several catalytic applications including H2 evolution, O2 evolution, CO2 reduction, N2 reduction, etc., However, in this section, only organic compound degradation has been discussed. I strongly recommend shedding light on as mentioned catalytic applications.
➢ Has been updated as advised
- Moreover, there are several branches of catalysis, such as photocatalysis, electrocatalysis, thermocatalysis, biocatalysis, etc. which are necessary to differentiate.
➢ Has been updated as advised
- Section 5.3 Pollutant removal is also a part of Catalysis, then why you separately discussed it? If you see carefully, in the catalysis section also, you discussed pollutant degradation. For example, “As a result, the corresponding TC removal and mineralization efficiencies increased to 91.66% and 45.04%, respectively.”
➢ Has been updated as advised
- A couple of sentences are hard to understand, make them accessible and understandable to the readers. For example, in the sentence “GO has drawn significant research interest due to its surface area resembling MOFs and the particular processing and handling advantages of dispersion and amphiphilicity in water provided by the oxygen-containing groups present” there are many ‘and’.
➢ Has been updated as advised
- There are several typos in the manuscript, correct them.
For example, (a) in the sentence ‘D’ of due should be a small case in the following sentence, “MOF-based GO has recently attracted attention Due to its resemblance to metal ions and organic binding linkers.
➢ Have been updated as advised
Comments on the Quality of English Language
The quality of English is good, but some minor typos need to be corrected.

Reviewer 4 Report
This article introduces the synthesis strategy of MOFs and their applications. MOFs represent clearly a sub-domain of MOFs research that has gained recently an increasing interest in many fields such as water or air purification, catalysis, energy storage and conversion, etc. This review is therefore timely and well-written. However, there are still important points, as well as mistakes, that need to be corrected prior to publication.
1. The review is organized into different parts, from the synthesis approaches to the different potential applications. A general discussion about the main advantages and current limitations of heterometallic MOFs in comparison with their homometallic counterparts. Synergestic points should be better discussed, for instance at the end of the paper.
2. Same comment for the challenges associated to a fine characterization of the presence of the two metals; a summary of the main recent progresses and remaining challenges of the usual characterization techniques should be given including recent publications.
3. When it deals with catalysis, authors should better explain how MOFs have their unique advantages in these three aspects and provide some examples.
4. Some examples on applications of MOFs could be updated, such as J. Mater. Chem. A, 2020, 8, 11933–11937; J. Mater. Chem. B, 2023, 11, 6335–6345; J. Mater. Chem. B, 2023, 11, 5693–5714; Dalton Trans, 2023, 52, 6226 – 6238 and J. Mol. Struct. 1291(2023)136074.
5. The quality of all figs should be improved, especially Fig 3,
6. A sub-section should be created to discuss the intrinsic and specific properties/features of MOF that makes them suitable to be used as super-capacitors
above mention
Author Response
- The review is organized into different parts, from the synthesis approaches to the different potential applications. A general discussion about the main advantages and current limitations of heterometallic MOFs in comparison with their homometallic counterparts. Synergetic points should be better discussed, for instance at the end of the paper.
- Has been updated as advised.
- Same comment for the challenges associated to a fine characterization of the presence of the two metals; a summary of the main recent progresses and remaining challenges of the usual characterization techniques should be given including recent publications.
- Has been updated as advised.
- When it deals with catalysis, authors should better explain how MOFs have their unique advantages in these three aspects and provide some examples.
- Has been updated as advised.
- Some examples on applications of MOFs could be updated, such as J. Mater. Chem. A, 2020, 8, 11933–11937; J. Mater. Chem. B, 2023, 11, 6335–6345; J. Mater. Chem. B, 2023, 11, 5693–5714; Dalton Trans, 2023, 52, 6226 – 6238 and J. Mol. Struct. 1291(2023)136074.
- The review paper focuses on MOFs/GO composites only not MOF.
- The quality of all figs should be improved, especially Fig 3,
- Has been updated as advised.
- A sub-section should be created to discuss the intrinsic and specific properties/features of MOF that makes them suitable to be used as super-capacitors
- Has been updated as advised.

Round 2
Reviewer 1 Report
The revised review has substantially enhanced the quality of the presentation and content of MOF-supported GO. However, the authors have largely overlooked the suggestions provided in the first revision. I strongly advise revisiting the review and incorporating the suggested data before considering it for publication. These are the suggestions I provided in my first revision.
1. Authors are recommended to provide a history and background of GO-MOFs in the manuscript to give more visibility for the review and a comprehensive focus for the readers.
2. Authors are recommended to compare the performance of early and recent pristine MOFs, such as SIFSIX and ALF. Present a comprehensive comparative table in the review. Also, authors are recommended to cite the below recent state-of-the-art MOFs that align with this topic. (Sci. Adv. 8, eade1473 (2022), J. Am. Chem. Soc. 2023, 145, 11643−11649, and J. Am. Chem. Soc. 2023, 145, 9850−9856).
3. Although this review briefly covers MOF and GO composite materials, it inadequately mentions future directions. Revisit the manuscript and explicitly mention the future of GO material for energy applications. Authors are strongly advised to revise the conclusion of the manuscript, which currently resembles an introduction. Provide detailed future directions along with concise achievements in the field of MOF-GO composite materials.
Here are some new suggestions:
1. The authors should correct the title of Figure-1 as it does not depict the structure of MOFs. Instead, it represents the schematic representation of MOF synthesis procedure.
2. The authors should cite and acknowledge the respective papers for Figure-2, Figure-3, and Figure-8, from where they have copied or adopted the images. The synthesis of graphene and GO is well-documented, so it is advisable to cite the papers whenever data or images are used properly.
3. The text in Figure-8 is difficult to read. Therefore, the authors should present a clearer visual figure in the manuscript.
Author Response
Has been updated.

Reviewer 2 Report
The reviewer suggestions were followed and the questions answered. Moreover, the authors inserted additional information in the manuscript, increasing its relevance.
Author Response
Thank you,
Reviewer 3 Report
The manuscript can be accepted in its current form. I would like to make a note here that although the revision has been made to reviewers' comments by the authors, I'm disappointed with the way the authors wrote the responses. While writing responses to the reviewer's comments, the author should mention what change they have made to particular comments, where these corrections have been made (section, line), how many new references were cited, what/where the action was taken to grammar/typos, etc. Instead, the authors simply wrote, "Has been updated as advised."
Author Response
Thank you,
I apologize and, in the future, I will respond with details.
We have revised for you and put your responses to the correct reviewer.
Author's Notes
Please finds the attachment.
- In the abstract, it has been mentioned that “In this review paper, an effort has been made to outline recent research on MOFs/GO materials features, advantages, properties, synthesis techniques, and applications in a variety of industries……”
However, you have only reviewed MOFs/GO for several applications, not for industrial applications. Make it clear and correct it if required.
- Has been updated to be (different applications). different applications, including supercapacitors, gas separation and storage, water purification, sensing, catalysis, and biomedical. (line 21-22).
- In this review, the progress of MOFs/GO has been carried out for several applications such as supercapacitors and so on. Nowadays, MOFs have received significant attention for several Biomedical applications such as disease diagnosis and drug delivery. I recommend the addition of the biological application of MOFs to section 5.
- ➢ Has been updated, and added new application Batteries (line 825) and Biomedical (line 867)
- In section 5 the applications of Metal-Organic Frameworks/Graphene Oxide (MOFs/GO) have been discussed. I recommend the addition of a new table that summarizes section 5 including materials, their applications, the results/output, reference, etc. Such a table will be helpful for the readers to understand the review more conveniently.
- ➢ Has been added new table (Table 2 summarizes MOFs-GO applications, including composite materials and the output) (line 903).
- I couldn’t see the reprint statement for the figures in their caption. If the figures are taken from any other articles, it is necessary to make a reprint statement and cite the article.
All Figures and Schemes, I made them myself and used the references to illustrate the process of the methods, except Scheme 1, which has copyrights. I received the permission through email because it was not available on RightsLink to obtain printable license permission. I uploaded the copyright and sent it as well through email to the assistant editor (Alexa Li).
- Section ‘5.5 Catalysis’ is too briefly discussed. MOFs are quite famous materials for several catalytic applications including H2 evolution, O2 evolution, CO2 reduction, N2 reduction, etc., However, in this section, only organic compound degradation has been discussed. I strongly recommend shedding light on as mentioned catalytic applications.
- ➢ Has been updated as advised, (Catalysis line 736, Electrocatalysis line 755, and Photocatalysis line 790)
- Moreover, there are several branches of catalysis, such as photocatalysis, electrocatalysis, thermocatalysis, biocatalysis, etc. which are necessary to differentiate.
- Has been updated as advised, (Electrocatalysis line 755 and Photocatalysis line 790).
- Section 5.3 Pollutant removal is also a part of Catalysis, then why you separately discussed it? If you see carefully, in the catalysis section also, you discussed pollutant degradation. For example, “As a result, the corresponding TC removal and mineralization efficiencies increased to 91.66% and 45.04%, respectively.”
- Has been changed to another application (Water purification, line 668).
- A couple of sentences are hard to understand, make them accessible and understandable to the readers. For example, in the sentence “GO has drawn significant research interest due to its surface area resembling MOFs and the particular processing and handling advantages of dispersion and amphiphilicity in water provided by the oxygen-containing groups present” there are many ‘and’.
- I reviewed the manuscript and changed many sentences to make them accessible and understandable. For example (GO has attracted significant research interest Because it contains a large surface area (2630 m2/g), similar to graphene, and the unique processing and handling benefits of amphiphilicity and dispersibility in water provided by the present oxygen-containing groups) (lines 168-169).
- There are several typos in the manuscript, correct them.
- I reviewed the manuscript and changed the incorrect typos.
For example, (a) in the sentence ‘D’ of due should be a small case in the following sentence, “MOF-based GO has recently attracted attention Due to its resemblance to metal ions and organic binding linkers.
➢ For example (Graphene oxide (GO) has attracted significant research interest due to its similar surface area to MOFs) (line 13-14)
Comments on the Quality of English Language
The quality of English is good, but some minor typos need to be corrected.

Reviewer 4 Report
the sentences and formation should be checked and revised
work
Author Response
The sentences and formation have been reviewed.